# DeepZero: Scaling Up Zeroth-Order Optimization for Deep Model Training

**Aochuan Chen**[†,⋆]  **Yimeng Zhang**[†,⋆]  **Jinghan Jia**[†]  **James Diffenderfer**[‡]  **Jiancheng Liu**[†]
**Konstantinos Parasyris**[‡]  **Yihua Zhang**[†]  **Zheng Zhang**[§]  **Bhavya Kailkhura**[‡]  **Sijia Liu**[†]
[†]Michigan State University, [‡]Lawrence Livermore National Laboratory, [§]UC Santa Barbara
[⋆]Equal contributions

## Abstract

Zeroth-order (ZO) optimization has become a popular technique for solving machine learning (ML) problems when first-order (FO) information is difficult or impossible to obtain. However, the scalability of ZO optimization remains an open problem: Its use has primarily been limited to relatively small-scale ML problems, such as sample-wise adversarial attack generation. To our best knowledge, no prior work has demonstrated the effectiveness of ZO optimization in training deep neural networks (DNNs) without a significant decrease in performance. To overcome this roadblock, we develop *DeepZero*, a principled ZO deep learning (DL) framework that can scale ZO optimization to DNN training from scratch through three primary innovations. *First*, we demonstrate the advantages of coordinate-wise gradient estimation (CGE) over randomized vector-wise gradient estimation in training accuracy and computational efficiency. *Second*, we propose a sparsity-induced ZO training protocol that extends the model pruning methodology using only finite differences to explore and exploit the sparse DL prior in CGE. *Third*, we develop the methods of feature reuse and forward parallelization to advance the practical implementations of ZO training. Our extensive experiments show that DeepZero achieves state-of-the-art (SOTA) accuracy on ResNet-20 trained on CIFAR-10, approaching FO training performance for the first time. Furthermore, we show the practical utility of DeepZero in applications of certified adversarial defense and DL-based partial differential equation error correction, achieving 10-20% improvement over SOTA. We believe our results will inspire future research on scalable ZO optimization and contribute to advancing DL with black box. Codes are available at `https://github.com/OPTML-Group/DeepZero`.

## 1 Introduction

In the realm of machine learning (ML), optimization algorithms have played a crucial role in enabling the training of complex models, yielding unprecedented insights and predictive capabilities across diverse domains. Over the years, first-order (FO) gradient-based methods, such as stochastic gradient descent (SGD) and its variants (Gardner, 1984; Amari, 1993; Bottou, 2010; 2012), have become the default choice for model training. These methods rely on gradient information to iteratively update model parameters, aiming to minimize a given loss function. Nonetheless, several practical settings exist where FO gradient information is either unavailable or infeasible to compute, calling for alternative strategies. Zeroth-order (ZO) optimization (Flaxman et al., 2005; Shamir, 2013; Ghadimi & Lan, 2013; Nesterov & Spokoiny, 2015; Duchi et al., 2015; Liu et al., 2018b; Ilyas et al., 2018b; Zhang et al., 2024) has emerged as a promising approach to address these challenges, as it leverages finite differences of function values to estimate gradients, rather than requesting explicit gradient information. Therefore, with minor modifications to FO algorithms, ZO optimization can be applied to various real-world circumstances where FO gradients are difficult to obtain. For example, in disciplines like physics and chemistry, ML models may interact with intricate simulators or experiments where the underlying systems are non-differentiable (Thelen et al., 2022; Tsaknakis et al., 2022; Louppe et al., 2019; Abreu de Souza et al., 2023; Baydin et al., 2020). Additionally, black-box learning scenarios often arise when deep learning (DL) models are integrated with third-party APIs, such as adversarial attack and defense against black-box DL models (Chen et al., 2017; Ilyas et al., 2018a; Zhang et al., 2022c; Verma et al., 2023) and black-box prompt learning

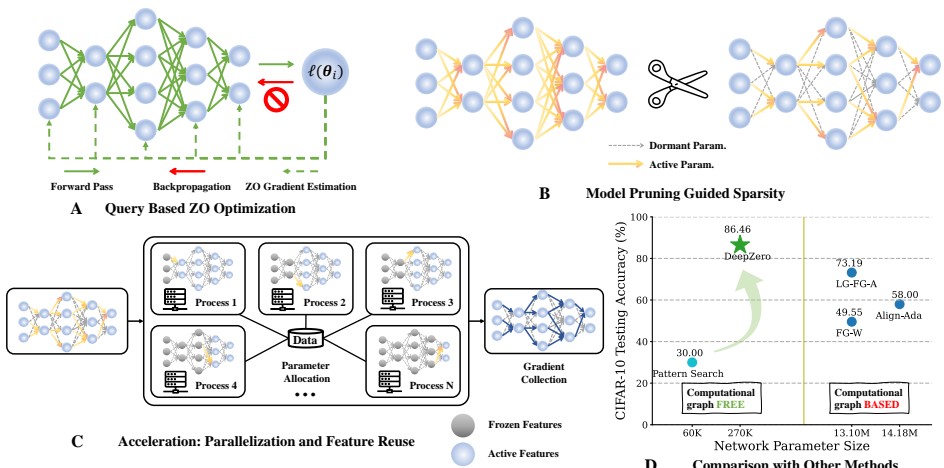

Figure 1: Overview of our DeepZero framework. **A:** ZO gradient estimation via model queries (Sec. 3). **B:** Model pruning guides gradient sparsity (Sec. 4). **C:** Acceleration by parallelization and feature reuse (Sec. 5). **D:** DeepZero comparison with the computational graph free baseline Pattern Search (Chiang et al., 2023) and computational graph dependent methods without BP, Align-Ada (Boopathy & Fiete, 2022), LG-FG-A and FG-W (Ren et al., 2022), on CIFAR-10.

for language-model-as-a-service (Diao et al., 2022; Sun et al., 2022). Furthermore, the principled backpropagation (BP) mechanism (Amari, 1993; Rumelhart et al., 1995) for calculating FO gradients may also not be supported when implementing DL models on hardware systems (Gu et al., 2021b; Tavanaei et al., 2019; Greengard, 2020; Jabri & Flower, 1992; Gu et al., 2021c). In addition to ZO optimization, another relevant research direction in the field of DL focuses on developing biologically-plausible, BP-free methods. Examples include forward gradient-based methods (Ren et al., 2022; Baydin et al., 2022; Silver et al., 2021; Belouze, 2022), greedy layer-wise learning (Nøkland & Eidnes, 2019), and Hebbian learning (Isomura & Toyoizumi, 2018; Moraitis et al., 2022). However, these techniques require access to computational graphs and are highly dependent on the used DL software frameworks and/or model architectures. In contrast, ZO optimization solely relies on model queries and is free of computational graphs utilized. As a result, ZO optimization has broad applicability to DL problems that involve black-box query-only components. Despite the promise of ZO optimization, scalability bottlenecks hinder its application in medium or large-scale DNN training (Wang et al., 2017; Liu et al., 2018b; Ohta et al., 2020; Cai et al., 2021; Zhang et al., 2022c). As problem dimensionality increases, the accuracy and efficiency of traditional ZO methods deteriorate. This is because ZO finite difference-based gradient estimates are biased estimators of FO gradients, and the bias becomes more pronounced in higher-dimensional spaces (Liu et al., 2018b; Cai et al., 2021; Balasubramanian & Ghadimi, 2018).

These challenges motivate the central question addressed in this work: *(Q) How to scale up ZO optimization for training deep models?* To address **(Q)**, we propose a novel framework, 'DeepZero', which infuses novel model pruning and parallel computing techniques to scale up ZO DNN training (see **Fig. 1** for a schematic overview). Our **main contributions** are summarized below.

❶ We show that deterministic coordinate-wise gradient estimation (CGE) outperforms vector-wise randomized gradient estimation (RGE) in both accuracy and computation efficiency when scaling to deep model training. Further, CGE becomes increasingly advantageous as model depth increases.

❷ We show that sparsity is crucial for realizing model training via CGE with finite differences. In contrast to prior work, we find that sparsity for black-box models can be obtained for 'free' by extending the current pruning-at-initialization technique to the ZO learning paradigm. The established synergy between pruning and CGE presents a promising avenue for efficient ZO training of DNNs.

❸ We identify the parallelization-fit property inherent to CGE-based ZO optimization and propose a novel *forward parallelization* method based on this property. Our framework enables *feature reuse* in deep learning, which further accelerates parallel training by eliminating redundant computations.

❹ We introduce our proposed ZO deep model training framework, 'DeepZero'. To demonstrate the empirical superiority of DeepZero, we conduct extensive experiments on both standard image classification benchmarks and real-world black-box DL applications. For example, when employing

DeepZero to train a ResNet20 on CIFAR-10, we obtain $86.94\%$ testing accuracy, the best reported in the literature of gradient-free model training. We also exemplify the vast potential and practical impact of DeepZero in two real-world DL tasks: black-box defense for adversarial robustness (Zhang et al., 2022c) and physics-informed DL with solver-in-the-loop (Um et al., 2020).

To clarify, our work aims to extend the scalability of ZO optimization for DL applications, addressing cases where FO optimization becomes challenging or infeasible. Yet, it is essential to note that the proposed advancements in ZO training are not intended to overcome the ultimate scalability challenges to train deep networks at any scale.

## 2 RELATED WORK

**Classical gradient-free optimization.** Early research efforts can be broadly categorized into two groups: direct search-based methods (DSMs) and model-based methods (MBMs) (Wright et al., 1999; Conn et al., 2009; Rios & Sahinidis, 2013; Larson et al., 2019). DSMs include techniques like coordinate (Fermi, 1952) and pattern search (Torczon, 1991) methods and the Nelder-Mead simplex method (Nelder & Mead, 1965). MBMs consist of model-based descent (Bortz & Kelley, 1998) and trust region (Conn et al., 2000) methods. Evolutionary optimization offers a generic population-based gradient-free computational framework including genetic algorithms (Grefenstette, 1993) and particle swarm optimization (Vaz & Vicente, 2009). Bayesian optimization (Shahriari et al., 2015; Eriksson et al., 2019) has garnered recent attention by using a Gaussian process (GP) to fit a black-box objective function and estimate an optimization solution. However, acquiring an accurate GP is computationally intensive.

**Zeroth-order optimization.** In contrast to classical gradient-free methods, ZO optimization approximates gradients using finite differences, simplifying implementation by minimizing modifications of FO gradient-based algorithms. Like FO methods, ZO enjoys provable convergence guarantees (Nesterov & Spokoiny, 2017; Duchi et al., 2015; Liu et al., 2020a). ZO optimization has gained significant attention for its success in solving various emerging ML problems (Ghadimi & Lan, 2013; Nesterov & Spokoiny, 2015; Flaxman et al., 2005; Duchi et al., 2015). Examples include adversarial attack and defense (Chen et al., 2017; Tu et al., 2019; Ye et al., 2018; Ilyas et al., 2018a; Zhang et al., 2022c; Verma et al., 2023; Zhao et al., 2019; Hogan & Kailkhura, 2018; Shu et al., 2022), model-agnostic contrastive explanation (Dhurandhar et al., 2019), visual prompting for transfer learning (Tsai et al., 2020), computational graph unrolling (Vicol et al., 2023), automated ML (Gu et al., 2021a; Wang et al., 2022), policy search in reinforcement learning (Vemula et al., 2019), network resource management (Liu et al., 2018b), ML-based scientific workflow optimization (Tsaknakis et al., 2022), and on-chip learning (Gu et al., 2021b). Despite ZO's successes in solving ML problems, its application has been limited to relatively small scales. For instance, ZO optimizers used for generating adversarial attacks, contrastive explanations, and visual prompts only operate in the input parameter space, which has the dimension of a single input example. Some acceleration techniques have been developed to improve ZO performance in larger problems, such as using historical information to enhance a ZO gradient estimator (Meier et al., 2019; Cheng et al., 2021), and exploiting gradient sparsity to reduce ZO dependence on problem size (Wang et al., 2017; Cai et al., 2022; 2021; Balasubramanian & Ghadimi, 2018; Ohta et al., 2020; Gu et al., 2021b). While gradient sparsity has been used to improve scalability (Bartoldson et al., 2023), we propose an advanced strategy that leverages model pruning techniques to identify and exploit sparsity in neural network parameters effectively. Our approach is less restrictive than traditional gradient sparsity assumptions and allows for greater flexibility in selecting what to prune. To the best of our knowledge, no prior work has demonstrated the practicality of scalable ZO optimization for deep model training without significant performance loss compared to the FO counterpart.

**DL without backpropagation.** Forward gradient learning (Baydin et al., 2022; Ren et al., 2022; Silver et al., 2021; Belouze, 2022), which builds upon the forward-mode automatic differentiation (AD) capabilities of current DL software frameworks, does not rely on finite differences to approximate FO gradients like ZO optimization. Instead, it relies on forward-mode AD to calculate a forward (directional) gradient. This gradient is obtained by projecting the FO gradient onto a direction vector and is an unbiased estimator of the FO gradient (Baydin et al., 2022). In contrast, ZO gradient estimation based on finite differences is biased (Duchi et al., 2015; Liu et al., 2020a). However, one main limitation of forward gradient learning is that it requires full access to AD software and the deep model, making it impractical for solving black-box DL problems. Recent advances in (Ren et al., 2022) further improved the scalability of forward gradient learning by using finer-level model information to design architecture-specific local objective functions. Other BP-free DL methods are

motivated by seeking a biological interpretation of DL but share similar limitations with forward gradient learning. Some examples include greedy layer-wise learning (Nøkland & Eidnes, 2019), input-weight alignment for wide neural networks in the neural tangent kernel (NTK) regime (Boopathy & Fiete, 2022), the Forward-Forward algorithm (Hinton, 2022), Hebbian Learning (Isomura & Toyoizumi, 2018; Moraitis et al., 2022), and synthetic gradients (Jaderberg et al., 2017).

# 3 ZO OPTIMIZATION THROUGH FUNCTION VALUE-BASED GRADIENT ESTIMATION: RANDOMIZED OR COORDINATE-WISE?

We now introduce the ZO optimization setup and discuss two ZO gradient estimation schemes: deterministic coordinate-wise gradient estimation (**CGE**) and randomized vector-wise gradient estimation (**RGE**). We will demonstrate the advantage of CGE over RGE for DNN training. This inspires further improvements for scaling CGE-based ZO optimization.

**ZO optimization and gradient estimation.** Let $\ell(\boldsymbol{\theta})$ denote a **loss function** that we want to minimize over the **optimization variables** $\boldsymbol{\theta} \in \mathbb{R}^d$ (*e.g.*, model parameters of a neural network). The ZO optimizer interacts with the objective function $\ell$ only by submitting inputs (*i.e.*, realizations of $\boldsymbol{\theta}$) and receiving the corresponding function values. It slightly modifies the commonly-used first-order (FO) gradient-based algorithm by approximating the FO gradient through function value-based gradient estimates (Liu et al., 2020a). This is essential when explicit differentiation is difficult due to the black-box nature of the loss function (Zhang et al., 2022c; Liu et al., 2020b; Chen et al., 2017), or when explicit differentiation is undesired due to concerns about energy efficiency (Gu et al., 2021b; Liu et al., 2018a). RGE (Nesterov & Spokoiny, 2017; Ghadimi & Lan, 2013; Duchi et al., 2015; Spall, 1992) and CGE (Kiefer & Wolfowitz, 1952; Lian et al., 2016; Berahas et al., 2022) are two commonly-used gradient estimators based on finite differences of $\ell$. RGE acquires finite differences via random perturbations of $\boldsymbol{\theta}$ while CGE uses deterministic coordinate-wise perturbations of $\boldsymbol{\theta}$ (Liu et al., 2020a). Their formal definitions are given by

$$\text{(RGE)} \ \hat{\nabla}_{\boldsymbol{\theta}}\ell(\boldsymbol{\theta}) = \frac{1}{q}\sum_{i=1}^{q}\left[\frac{\ell(\boldsymbol{\theta}+\mu\mathbf{u}_i)-\ell(\boldsymbol{\theta})}{\mu}\mathbf{u}_i\right]; \ \text{(CGE)} \ \hat{\nabla}_{\boldsymbol{\theta}}\ell(\boldsymbol{\theta}) = \sum_{i=1}^{d}\left[\frac{\ell(\boldsymbol{\theta}+\mu\mathbf{e}_i)-\ell(\boldsymbol{\theta})}{\mu}\mathbf{e}_i\right], \ (1)$$

where $\hat{\nabla}_{\boldsymbol{\theta}}\ell$ denotes an estimation of the FO gradient $\nabla_{\boldsymbol{\theta}}\ell$ with respect to $\boldsymbol{\theta}$. In (RGE), $\mathbf{u}_i$ denotes a randomized perturbation vector, *e.g.*, drawn from the standard Gaussian distribution $\mathcal{N}(\mathbf{0}, \mathbf{I})$, $\mu > 0$ is a perturbation size (*a.k.a.* smoothing parameter), and $q$ is the number of random directions used to acquire finite differences. In (CGE), $\mathbf{e}_i$ denotes a standard basis vector, and $\frac{\ell(\boldsymbol{\theta}+\mu\mathbf{e}_i)-\ell(\boldsymbol{\theta})}{\mu}$ provides the finite-difference estimation of the partial derivative of $\ell(\boldsymbol{\theta})$ at the $i$th coordinate $\boldsymbol{\theta}_i$. Finite difference approximations in (1) are motivated by *directional derivatives*. Take RGE (with $q = 1$) as an example. As $\mu \to 0$, finite difference in RGE converges to directional derivative $\ell'(\boldsymbol{\theta}) := \mathbf{u}^T\nabla_{\boldsymbol{\theta}}\ell(\boldsymbol{\theta}) = \lim_{\mu\to 0}\frac{\ell(\boldsymbol{\theta}+\mu\mathbf{u}_i)-\ell(\boldsymbol{\theta})}{\mu}$ of the function $\ell$ at the point $\boldsymbol{\theta}$ in the direction $\mathbf{u}$ (Urruty &

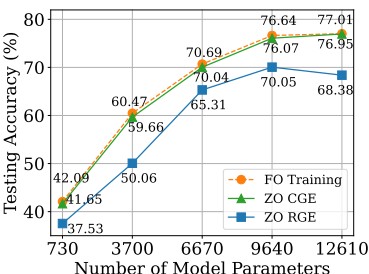

Figure 2: Performance comparison of training a simple CNN with varying numbers of parameters on CIFAR-10 using different training methods.

Lemaréchal, 1993). As a result, the expression $\ell'(\boldsymbol{\theta})\mathbf{u}$ yields $\mathbb{E}[\ell'(\boldsymbol{\theta})\mathbf{u}] = \mathbb{E}[(\mathbf{u}\mathbf{u}^T)\nabla_{\boldsymbol{\theta}}\ell(\boldsymbol{\theta})] = \nabla_{\boldsymbol{\theta}}\ell(\boldsymbol{\theta})$ (recall that $\mathbb{E}[\mathbf{u}\mathbf{u}^T] = \mathbf{I}$). This implies that $\ell'(\boldsymbol{\theta})\mathbf{u}$ is an *unbiased* gradient estimator of $\nabla_{\boldsymbol{\theta}}\ell(\boldsymbol{\theta})$ and its *biased* finite difference approximation is given by (1) (Duchi et al., 2015).

**RGE or CGE?** First, the *function query costs* for RGE and CGE differ, with RGE taking $O(q)$ queries and CGE taking $O(d)$ queries based on (1). Compared to CGE, RGE has the flexibility to specify $q < d$ to reduce the number of function evaluations. Despite the query efficiency, it remains uncertain whether RGE can deliver satisfactory accuracy when training a deep model from scratch. To this end, we undertake a preliminary investigation wherein we train a basic convolutional neural network (CNN) of different sizes on CIFAR-10, employing both RGE and CGE. To ensure a fair comparison in query complexity, we set the query number $q$ in RGE equal to the problem size $d$ used in CGE. **Fig. 2** presents the test accuracy of the learned CNN against the number of model parameters (equivalent to the number of model queries). Here the training recipe is specified by the FO SGD, the ZO RGE-based SGD, and the ZO CGE-based SGD. We observe that CGE can achieve test accuracy comparable to FO training and significantly outperforms RGE. This experiment highlights the superiority of CGE over RGE in terms of optimization accuracy even when the

latter uses $q = d$. This *accuracy merit* of CGE is particularly valuable when training more complex neural networks. In Appx. C, we provide a detailed analysis of the computational costs using CGE vs. RGE. The time cost relative to model depth is shown in Fig. A2. And Tab. A1 assesses gradient estimation time costs. We find that CGE demonstrates greater time efficiency than RGE. The computation efficiency loss of RGE is in that it needs to generate and integrate a $d$-dimension perturbation vector into the entire model at once every query. It is worth noting that the recent work (Malladi et al., 2024) suggested reducing the memory cost of RGE by storing only the random seed and regenerating the random vector when required. Yet, we argue that this may further hamper the computation efficiency of RGE due to the need for repeated generation. Based on the advantages of CGE over RGE in terms of both accuracy and computation efficiency, we choose CGE as the preferred ZO gradient estimator. However, query complexity of CGE is still a bottleneck, as it scales with model size $d$.

## 4    SPARSITY-ASSISTED ZO TRAINING: A PRUNING LENS AND BEYOND

One valuable property of CGE is the disentanglement of finite differences across coordinates, which suggests that reducing CGE's query complexity is aligned with pruning the model weights that are being optimized. With this in mind, we propose integrating ZO optimization with *pruned gradients* to design a more effective inductive bias for ZO deep model training. It is worth noting that the sparsity has been explored in several existing ZO optimization methods to improve the query efficiency of gradient estimation (Wang et al., 2017; Cai et al., 2022; 2021; Balasubramanian & Ghadimi, 2018; Ohta et al., 2020; Gu et al., 2021b). However, prior work suffered from two main limitations. Firstly, exact sparsity was assumed in the original FO gradients, which required an additional sparse learning method (such as LASSO (Wang et al., 2017)) to recover these sparse gradients from function queries. Secondly, it remains unclear how to optimize the sparsity pattern via a ZO oracle, as the existing method calls for overly heuristics-based pruning methods (e.g., random (Gu et al., 2021b) or magnitude (Ohta et al., 2020; Zhang et al., 2022b) pruning). Overly increasing sparsity ultimately limits optimization performance. In what follows, we propose a new pruning approach that relies only on model queries, enjoys computation efficiency, and can improve ZO optimization accuracy by inducing an appropriate gradient sparsity.

**ZO-GraSP: Model pruning via ZO oracle.** The compressibility of model weights for DL has been extensively studied (Han et al., 2015; Frankle & Carbin, 2018; Ma et al., 2021; Zhang et al., 2022a;b; Blalock et al., 2020; Tanaka et al., 2020; Lee et al., 2018; Wang et al., 2020; Su et al., 2020; Diffenderfer et al., 2021). For instance, the lottery ticket hypothesis (Frankle & Carbin, 2018) demonstrated that a randomly initialized, dense neural network contains a high-quality *sparse subnetwork*. However, current effective pruning methods incorporate model training as an intermediate step (Frankle & Carbin, 2018; Ma et al., 2021; Zhang et al., 2022a; Diffenderfer & Kailkhura, 2021). *Thus, they are not well-suited for finding sparsity via a ZO oracle.*

To address the above challenge, we draw inspiration from training-free pruning methods, known as pruning-at-initialization (Tanaka et al., 2020; Lee et al., 2018; Wang et al., 2020). Within this family, gradient signal preservation (GraSP) (Wang et al., 2020) is a method to identify the sparsity prior of DL through the gradient flows of a randomly-initialized network. While GraSP still requires the FO and second-order derivative information, we can estimate these derivatives using only function queries to design the ZO version of GraSP (termed **ZO-GraSP**). Specifically, GraSP (Wang et al., 2020) assigns pruning scores (denoted by $\mathbf{S}$) to model initialization $\boldsymbol{\theta}$. These scores reflect the change in gradient flow after pruning the weights:

$$\mathbf{S} = -\boldsymbol{\theta} \odot (\mathbf{Hg}), \ \ \mathbf{H} = \nabla^2_{\boldsymbol{\theta},\boldsymbol{\theta}}\ell(\boldsymbol{\theta}), \ \mathbf{g} = \nabla_{\boldsymbol{\theta}}\ell(\boldsymbol{\theta}), \tag{2}$$

where recall that $\ell$ is the loss function of model training, $\odot$ denotes the entry-wise multiplication, and $\mathbf{Hg}$ represents the Hessian-gradient product. Using the ZO learning paradigm, we can first approximate the Hessian-gradient product as the finite difference between two gradients (*i.e.*, $\nabla_{\boldsymbol{\theta}}\ell(\boldsymbol{\theta} + \mu\mathbf{g})$ and $\nabla_{\boldsymbol{\theta}}\ell(\boldsymbol{\theta})$), in the direction $\mathbf{g}$ with the smoothing parameter $\mu$. Second, we replace the FO gradient $\nabla_{\boldsymbol{\theta}}\ell$ with the ZO gradient estimate $\hat{\nabla}_{\boldsymbol{\theta}}\ell$ given in (1). Combining this yields ZO-GraSP:

$$\hat{\mathbf{S}} := -\boldsymbol{\theta} \odot \frac{\hat{\nabla}_{\boldsymbol{\theta}}\ell(\boldsymbol{\theta} + \mu\hat{\mathbf{g}}) - \hat{\nabla}_{\boldsymbol{\theta}}\ell(\boldsymbol{\theta})}{\mu}. \tag{3}$$

In practice, we found that the pruning mask determined by ranking the entries in $\hat{\mathbf{S}}$ is resilient to the ZO gradient estimation error. Therefore, we utilize RGE with a relatively small number of queries ($q < d$) to implement ZO-GraSP. This reduces the function query cost without compromising pruning performance; see **Tab. A2** and **Tab. A3** for empirical justifications. Our results show

that ZO-GraSP significantly outperforms random pruning and yields pruned models with accuracy comparable to FO-GraSP.

**Integrating sparsity with CGE.** As finite differences in CGE (1) are decomposable over weights, it is easy to incorporate sparsity into CGE. To retain the accuracy benefits of training dense models, we incorporate gradient sparsity (in CGE) rather than weight sparsity. This ensures that we train a dense model in the weight space, rather than training a sparse model where the sparsity determined by ZO-GraSP is directly applied. Let $\mathcal{S}_{\text{ZO-GraSP}}$ be the coordinate set of unpruned model weights found by ZO-GraSP. The sparsity-induced CGE is given by

$$\hat{\nabla}_{\boldsymbol{\theta}}\ell(\boldsymbol{\theta}) = \sum_{i \in \mathcal{S}_{\text{ZO-GraSP}}} \left[ \frac{\ell(\boldsymbol{\theta} + \mu \mathbf{e}_i) - \ell(\boldsymbol{\theta})}{\mu} \mathbf{e}_i \right]. \qquad \text{(Sparse-CGE)}$$

It is clear that (Sparse-CGE) reduces the query complexity of the original CGE from $O(d)$ to $O(|\mathcal{S}_{\text{ZO-GraSP}}|)$, where $|\mathcal{S}_{\text{ZO-GraSP}}|$ denotes the cardinality of the coordinate set $\mathcal{S}_{\text{ZO-GraSP}}$. There may exist two direct methods for integrating (Sparse-CGE) into ZO optimization. $\mathcal{M}_1$: This method involves *alternating* between ZO-GraSP and CGE-based ZO optimization. At each iteration, $\mathcal{S}_{\text{ZO-GraSP}}$ is updated based on the model weights from the previous iteration and then used to construct (Sparse-CGE) for updating $\boldsymbol{\theta}$ at the current iteration. $\mathcal{M}_2$: This method involves performing *pruning before ZO training*. That is, ZO-GraSP is conducted at model initialization, and the resulting $\mathcal{S}_{\text{ZO-GraSP}}$ is applied to (Sparse-CGE) and kept fixed during training. **Both $\mathcal{M}_1$ and $\mathcal{M}_2$ have limitations.** $\mathcal{M}_1$ requires repeated calls to ZO-GraSP to update $\mathcal{S}_{\text{ZO-GraSP}}$, leading to a higher query cost for ZO model training. $\mathcal{M}_2$ addresses the query complexity by performing ZO-GraSP before training, but it can only produce a smaller model after training. It is known that heavily-pruned models suffers from performance degradation (*e.g.*, 95% sparse model in Tab. A2 in Appx. D). Thus, it is nontrivial to integrate ZO-GraSP with ZO training due to the requirement of balancing query efficiency and training effectiveness. To address this, we propose **ZO-GraSP-oriented dynamic sparsity pattern**, which leverages ZO-GraSP to determine layer-wise pruning ratios (LPRs) that can capture DNN compressibility. This approach shares a similar essence with smart ratio introduced in (Su et al., 2020). Specifically, we acquire LPRs from ZO-GraSP at randomly initialized weights prior to ZO training, which is query-efficient like $\mathcal{M}_2$. However, unlike $\mathcal{M}_2$, LPRs allow for random shuffling of sparse gradient positions in $\boldsymbol{\theta}$ only if these LPRs are obeyed. This allows us to mimic $\mathcal{M}_1$ to alternate between model weight updates and $\mathcal{S}_{\text{ZO-GraSP}}$ updates, with the latter achieved by LPR-guided randomly updated sparsity patterns. Thus, ZO optimization can train the dense model using iteratively-updated (Sparse-CGE) with LPRs-guided dynamic sparsity patterns. Overall, our proposal has the query efficiency of $\mathcal{M}_2$ with the training effectiveness of $\mathcal{M}_1$, resulting in a balanced integration of ZO-GraSP into ZO training. We summarize the algorithmic pipeline in **Algorithm 1** in Appx. E, where CGE and ZO-GraSP-oriented dynamic sparsity pattern are clearly described in a unified framework. We also refer readers to Appx. E for more explanation and comparisons with $\mathcal{M}_1$ and $\mathcal{M}_2$. We provide a convergence rate analysis in Appx. A.

## 5 IMPROVING SCALABILITY: FEATURE REUSE & FORWARD PARALLEL

We investigate two characteristics of ZO training that can further enhance implementation scalability: *feature reuse* and *forward parallelization*. The former disentangles intermediate features from weight perturbations, while the latter uses the finite-difference nature in CGE to enable scalable distributed implementation.

**Reusing intermediate features.** As shown in (1), CGE perturbs each parameter element-wise. Thus, one can reuse the feature immediately preceding the perturbed layer and carry out the remaining forward pass operations instead of starting from the input layer, as illustrated in **Fig. 1**. The above characteristic of CGE-based model training is referred to as '**feature reuse**'. More concretely, let $f_{\boldsymbol{\theta}}(\mathbf{x})$ be a deep model with parameters $\boldsymbol{\theta}$ and input $\mathbf{x}$. We can express $f_{\boldsymbol{\theta}}(\mathbf{x})$ as a multi-layer composite function

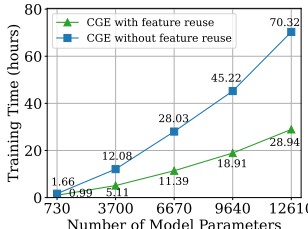

Figure 3: Computation cost of CGE-based ZO training w/ feature reuse vs. w/o feature reuse. The setup follows Fig. A2.

$$f_{\boldsymbol{\theta}}(\mathbf{x}) = f_{\boldsymbol{\theta}_{>l}}(\mathbf{z}_l) = \underbrace{f_{\boldsymbol{\theta}_L} \circ f_{\boldsymbol{\theta}_{L-1}} \circ \cdots \circ f_{\boldsymbol{\theta}_{l+1}}}_{f_{\boldsymbol{\theta}_{>l}}(\cdot)} \circ \underbrace{f_{\boldsymbol{\theta}_l} \circ \cdots \circ f_{\boldsymbol{\theta}_1}(\mathbf{x})}_{\mathbf{z}_l = f_{\boldsymbol{\theta}_{1:l}}(\mathbf{x})}, \qquad (4)$$

where $f_{\boldsymbol{\theta}_l}$ denotes the model's $l$th layer, $L$ is the total number of model layers, and $\circ$ is the function composition operation. Based on (4), if coordinate-wise weight perturbations are applied to the

$(l + 1)$th layer and its subsequent layers (*i.e.*, $\boldsymbol{\theta}_{>l}$), the model's outputs corresponding to these perturbed weights can be efficiently obtained by keeping the intermediate features up to the $l$th layer (*i.e.*, $\mathbf{z}_l$) intact. This efficiency becomes more pronounced when perturbing the parameters of deeper layers (*i.e.*, for a larger $l$). **Fig. 3** compares the runtime of CGE-based ZO training with and without feature reuse. Empirically, CGE with feature reuse exhibits a $2\times$ reduction in training time.

**Parallelization of coordinate-wise finite differences.** CGE enables parallelization of model training due to its alignment of parameter perturbations with forward passes. If there exist $M$ processes (across multiple GPUs), we can decompose CGE (1) based on its parameter coordinates yielding

$$\hat{\nabla}_{\boldsymbol{\theta}}\ell(\boldsymbol{\theta}) = \sum_{i=1}^{M} \hat{\mathbf{g}}_i, \ \ \hat{\mathbf{g}}_i := \sum_{j \in \mathcal{S}_i} \left[ \frac{\ell(\boldsymbol{\theta} + \mu\mathbf{e}_j) - \ell(\boldsymbol{\theta})}{\mu}\mathbf{e}_j \right], \tag{5}$$

where $\mathcal{S}_i$ is the set of active parameters assigned to process $1 \leq i \leq M$. Hence, each process can take $|\mathcal{S}_i|$ forward passes. This decoupling property enables scaling forward passes via distributed machines, which can significantly improve training speed. We refer to this parallelization for finite differences as '**forward parallelization**'. It is worth noting that forward parallelization is different from the conventional data parallelization used for FO distributed training (Goyal et al., 2017; You et al., 2018). Our method avoids any performance loss that may result from data parallelization using an overly large batch size, which can cause the optimizer to get stuck in suboptimal local minima due to a lack of stochasticity.

## 6 EXPERIMENTS

In this section, we first train *ResNet-20* for standard image classification on *CIFAR-10*, demonstrating scalability and generalization capability over existing gradient-free learning methods. Second, we apply DeepZero to enhance the robustness of a *black-box* DNN against adversarial attacks, where limited access to the model is available on the defender's end. Lastly, we leverage DeepZero to design a *physics-informed ML* system by incorporating a scientific PDE solver into the training loop for reducing numerical errors, highlighting its capability to address complex scientific problems.

### 6.1 IMAGE CLASSIFICATION TASK

**Experiment setup.** This study focuses on training ResNet-20 (with 270K parameters) on CIFAR-10 for image classification. We adopt SGD (stochastic gradient descent) as the FO training recipe, with a weight decay of $5 \times 10^{-4}$ and a momentum of $0.9$. The learning rate is $0.1$, governed by a cosine decay scheduler. In the ZO training scenario, we replace the FO gradient by (Sparse-CGE) with a smoothing parameter $\mu = 5 \times 10^{-3}$. When implementing ZO-GraSP (3), we set the query budget $q = 192$ and use the same $\mu$ as CGE. Unless specified otherwise, the weight sparsity ratio is chosen to be 90% and the specific sparsity patterns are determined by SR (Smart Ratio). When implementing DeepZero (**Algorithm 2**), we choose the number of epochs $T = 50$. Experiments are run on 4 NVIDIA V100 GPUs if not specified otherwise. We compare DeepZero with FO training and two SOTA BP-free training: Pattern Search (Chiang et al., 2023) and Input-Weight Alignment (Align-ada) (Boopathy & Fiete, 2022).

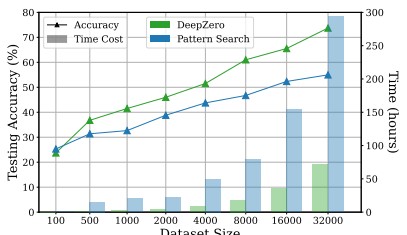

Figure 4: Comparison between DeepZero and FO training baselines on a ResNet-20 for CIFAR-10. We report the mean and standard deviation of 3 independent runs for each experiment.

**Comparison with FO training.** In **Fig. 4**, we compare the accuracy of DeepZero-trained ResNet-20 with two variants trained by FO recipes: (1) a dense ResNet-20 acquired through FO training and (2) a sparse ResNet-20 acquired through FO training under FO-GraSP sparsity pattern. As we can see, the accuracy gap still exists between (1) and the model trained with DeepZero in the sparsity regime of 80% to 99%. This highlights the challenge of ZO optimization for deep model training, where achieving high sparsity is desired to reduce the number of model queries in (Sparse-CGE) for scaling to ResNet-20. Notably, in the sparsity regime of 90%

Figure 5: Comparison of DeepZero and Pattern Search on ResNet-20 for CIFAR-10 with varying dataset sizes. All experiments are done on a single NVIDIA A6000 GPU.

to 99%, DeepZero outperforms (2), showcasing the superiority of gradient sparsity in DeepZero compared to weight sparsity (*i.e.*, directly training a sparse model). In Appx. F, we provide the DeepZero training trajectory (**Fig. A4**), performance vs. data batch setup (**Fig. A5**) and training time vs. GPU count (**Fig. A6**).

**Comparison with pattern search (Chiang et al., 2023).** In **Fig. 5**, we compare the accuracy and runtime cost of DeepZero with Pattern Search (Chiang et al., 2023) for deep model training. Pattern Search has been shown to achieve comparable test accuracy to SGD in low sample regimes, however, effectiveness as the number of data samples increases remains unclear. To investigate this, we conducted experiments using DeepZero (with 90% gradient sparsity) and pattern search on ResNet-20 with CIFAR-10, with varying dataset sizes from 100 to 32K. We maintained a fixed total epoch number of 40 for both methods to ensure a fair comparison. The results demonstrate DeepZero outperforms Pattern Search in all data regimes, except in the case of 100. Further, the improvement of DeepZero over pattern search (in both accuracy and efficiency) expands dramatically with increasing dataset sizes, indicating the superior scalability of DeepZero.

**Comparison with input-weight alignment (Boopathy & Fiete, 2022).** In **Tab. 1**, we present a comparison between DeepZero and the *Align-ada* approach (Boopathy & Fiete, 2022) for training neural networks without BP on CIFAR-10. While other BP-free training methods (Lillicrap et al., 2014; Nøkland, 2016; Baydin et al., 2022) exist, Align-ada stands out as it applies to training *wide*

Table 1: Performance of DeepZero vs. BP-free methods on a 8-layer CNN w/ different widths (Boopathy & Fiete, 2022).

| Method | DeepZero | | FA | | DFA | | Align-ada | |
|---|---|---|---|---|---|---|---|---|
| Width | 32 | 64 | 64 | 512 | 64 | 512 | 64 | 512 |
| Accuracy | 57.7 | **64.1** | 46.5 | 45.4 | 49.9 | 54.1 | 49.9 | 58.0 |
| Time (h) | 4.34 | 28.15 | 0.36 | 3.79 | 0.42 | 3.52 | 0.48 | 4.59 |

neural networks and achieves state-of-the-art performance on CIFAR-10, *e.g.*, surpassing methods such as feedback alignment (FA) (Lillicrap et al., 2014) and direct feedback alignment (DFA) (Nøkland, 2016). To ensure fairness, we apply DeepZero to the 8-layer CNN architecture from (Boopathy & Fiete, 2022) and compare performance with Align-ada at varying model widths. We note that the 512-width network was the widest model trained using Align-ada. In contrast, the largest width network we train with DeepZero is 64. Our results clearly show that DeepZero achieves significantly higher testing accuracy compared to Align-ada, even when training with narrower networks. This demonstrates that the improved performance of DeepZero is attributed to its inherent optimization advantages, rather than relying solely on the use of wider networks. Lastly, it is worth noting that Align-ada and other BP-free methods rely on access to computational graphs, making them efficient but unsuitable for black-box applications.

## 6.2    OTHER BLACK-BOX APPLICATIONS

**Black-box defense against adversarial attacks.** The black-box defense problem arises when the owner of an ML model is unwilling to share the model details with the defender against adversarial attacks (Zhang et al., 2022c; Verma et al., 2023). This poses a challenge for existing robustness enhancement algorithms (Madry et al., 2017; Cohen et al., 2019; Salman et al., 2020) that directly robustify white-box ML models using FO training. To overcome this challenge, ZO optimization was introduced in (Zhang et al., 2022c) to design a *white-box* defense operation given a *query-based black-box* image classifier. To address dimensionality challenges with ZO, ZO-AE-DS (Zhang et al., 2022c)

Table 2: CA (%) vs. $\ell_2$-norm based perturbation radius on ImageNet-10 using FO DS-based defense (FO-DS) (Salman et al., 2020), ZO-AE-DS (Zhang et al., 2022c), and our proposed DeepZero.

| ImageNet (10 classes) | | | |
|---|---|---|---|
| Radius $r$ | FO-DS | ZO-AE-DS | DeepZero |
| 0.0 | 89.33 | 63.60 | 86.02 |
| 0.25 | 81.67 | 52.80 | 76.61 |
| 0.5 | 68.87 | 43.13 | 61.80 |
| 0.75 | 49.80 | 32.73 | 43.05 |

2022c) introduces an autoencoder (AE) between the *white-box* denoised smoothing (DS) defense operation (to be learned) and the black-box image classifier. By merging AE's encoder with the black-box module, the dimension of ZO optimization is reduced; see **Fig. A7** in Appx. G for a schematic overview and (Zhang et al., 2022c) for details. The downside of ZO-AE-DS is poor scaling to high-resolution datasets (*e.g.*, ImageNet) due to the use of AE, which compromises the fidelity of the image input to the black-box image classifier and leads to inferior defense performance. In contrast, DeepZero can directly learn the defense operation integrated with the black-box classifier, without needing AE. **Tab. 2** compares the defense performance of DeepZero with FO defense (DS (Salman et al., 2020)) and ZO-AE-DS (Zhang et al., 2022c). To ensure a fair comparison, we used the same number of queries (1152) per gradient estimation. Following (Zhang et al., 2022c), we selected a 10-class subset of ImageNet as the training set. The AE is given by DnCNN (Zhang et al., 2017) and the

black-box classifier is specified by ResNet-50. Defense performance is evaluated by certified accuracy (CA), following the setup of (Salman et al., 2020; Zhang et al., 2022c). CA is defined using the $\ell_2$ norm-based input perturbation radius $r$, where a larger $r$ indicates a stronger adversarial threat. We refer readers to Appx. G for more experiment details. **Tab. 2** highlights that DeepZero consistently outperforms ZO-AE-DS in terms of CA for all values of $r > 0$. It is important to note that when $r = 0$, CA is equivalent to the standard testing accuracy. This indicates that DeepZero excels over ZO-AE-DS not only in adversarial robustness but also in overall generalization performance.

**Simulation-coupled DL for discretized PDE error correction.** Numerical methods, while instrumental in providing physics-informed simulations, come with their own challenge: the discretization unavoidably produces numerical errors. DL has gained significant attention for addressing this error correction problem. The feasibility of training a corrective neural network through looping interactions with the iterative partial differential equation (PDE) solver, coined 'solver-in-the-loop' (**SOL**), has been demonstrated in (Um et al., 2020). While existing work focused on using or developing differentiable simulators for model training, we extend SOL by leveraging DeepZero, enabling its use with non-differentiable or black-box simulators. We name our method **ZO-SOL** and refer readers to **Fig. A8** in Appx. H for a schematic overview. In this

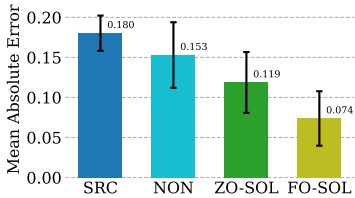

Figure 6: Average MAE of corrected low-fidelity simulations compared to high-fidelity simulations over 5 test simulations using different correction methods. Error bar is variance of MAE across 5 test simulations.

experimental framework, the goal is to correct the output of a low fidelity (*i.e.*, coarse mesh) simulation using a learned DNN so that the corrected simulation more closely aligns with output of a high fidelity (*i.e.*, fine mesh) simulation. In an effort to reduce the amount of correction required by DNN, this correction is applied at each simulation timestep and the corrected simulation state is provided to the simulator to compute the next timestep. For our experiments, we consider 2D unsteady wake flow fluid dynamics benchmark from (Um et al., 2020), in which there is a static rod around which fluid is flowing, and utilize PhiFlow (Holl et al., 2020) as the simulator. Additional details on the PDE, the solver-in-the-loop loss, and the DNN architecture are given in Appx. H. **Fig. 6** compares the test error correction performance of ZO-SOL (via DeepZero) with three differentiable approaches methods considered in (Um et al., 2020): SRC (low fidelity simulation without error correction), NON (non-interactive training out of the simulation loop using pre-generated low and high fidelity simulation data), and FO-SOL (FO training for SOL given a differentiable simulator). The simulation error for each method in **Fig. 6** is measured as the average across the five test simulations (with varying Reynold's numbers that were not utilized in the training set of simulation data). The error for each test simulation is computed as the mean absolute error (MAE) of the corrected simulation compared to the high fidelity simulation averaged across all simulation timesteps. We implement DeepZero for ZO-SOL following the setup used in the image classification task, except for choosing a 95% gradient sparsity. The ZO-SOL and FO-SOL use 16 unrolling steps in the loss function to allow the correction function to interact with the simulator during training. The results demonstrate that ZO-SOL achieved by DeepZero outperforms the SRC and NON baselines, and narrows the performance gap with FO-SOL, despite only having query-based access to the simulator. Comparing ZO-SOL with NON highlights the promise of ZO-SOL even when integrated with black-box simulators.

## 7    CONCLUSION

This paper introduces DeepZero, a framework designed to enhance the scalability of ZO optimization for deep network training. Specifically, DeepZero integrates coordinate-wise gradient estimation, ZO pruning-enabled gradient sparsity, feature reuse, and forward parallelization into a unified training pipeline. Leveraging these innovations, DeepZero showcases both efficiency and effectiveness in a wide range of applications, including image classification tasks and various practical black-box DL scenarios. While DeepZero has made remarkable progress in training DL models on datasets like ResNet-20 and CIFAR-10, it is important to acknowledge that scalability remains a significant challenge when dealing with even larger models and more extensive datasets. Future studies to accelerate ZO optimization for DL are necessary. Additionally, it is worthwhile to explore the applicability of DeepZero in other domains, such as digital twin applications that involve non-differentiable physical entities, and on-device training where the computational overhead of establishing computational graphs and backpropagation is not feasible. Lastly, we refer readers to Appx. I for a discussion of the broader impact of this paper.

ACKNOWLEDGMENT

We thank the U.S. Department of Energy via Lawrence Livermore National Laboratory under Contract DE-AC52- 07NA27344 and the LLNL-LDRD Program under Project No. 23-ER-030 (LLNL-CONF-849161) for their support.

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

APPENDIX

## A  REMARK ON CONVERGENCE RATE.

For a rigorous convergence analysis, we can perceive the sparsity-enhanced ZO training method introduced above as a specific instantiation within the broader framework of ZO stochastic coordinate descent (Lian et al., 2016). Thus, under necessary theoretical assumptions regarding Lipschitz continuity, gradient smoothness, and bounded gradient variance, we can obtain an upper bound for the convergence rate of our proposal in terms of the the gradient norm, serving as a stationarity-based convergence measure for non-convex optimization:

$$\frac{\sum_{k=0}^{K} \mathbb{E}|\nabla f(\boldsymbol{\theta}_k)|^2}{K} \leq O\left(\frac{1}{K(1-p)} + \sqrt{\frac{1}{K(1-p)}}\sigma\right), \tag{A1}$$

where $k$ represents the iteration index, $K$ signifies the total number of iterations, $O$ is used in the context of big O notation, $p$ denotes the gradient sparsity ratio, and $\sigma$ stands for the upper bound on the variance of the stochastic gradients. The above highlights that as $p$ increases, the convergence error of the proposed approach also grows. However, it's important to note that increased sparsity leads to a reduced number of function queries. Consequently, we can see a provable tradeoff between the convergence error and the query efficiency, aligning well with our understanding.

## B  THE SIMPLE CNN ARCHITECTURE CONSIDERED FOR TRAINING W/ CGE VS. RGE

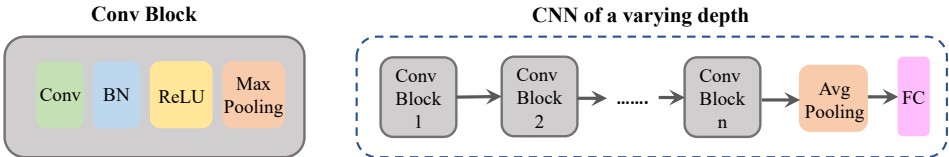

Figure A1: Illustration of the simple CNN considered with different depths.

As illustrated in **Fig. A1**, the depth-adjustable simple CNN architecture comprises multiple conv blocks, an average pooling layer, and a fully-connected layer. Each conv block consists of four distinct layers, including ① convolutional layer, ② batch normalization layer, ③ ReLU layer, ④ max pooling layer. In our experiments, the simple CNN networks are trained over a course of 50 epochs utilizing three distinct optimization strategies: FO (first-order) SGD (stochastic gradient descent) training, ZO RGE-based SGD training, and ZO CGE-based SGD training, together with a cosine learning rate scheduler is used.

## C  COMPUTATION TIME COMPARISON BETWEEN RGE AND CGE

We find that CGE also has a *computatoin efficiency merit* over RGE when training CNNs in **Fig. 2**. **Fig. A2** presents the computation time of different training methods against the model size and highlights that CGE is more computationally efficient than RGE. To examine the efficiency merit of CGE in detail, we dissect the computation of a single ZO gradient estimate into *four stages*: ① generation of directional vectors, *i.e.*, $\mathbf{u}_i$ or $\mathbf{e}_i$ in (1), ② perturbation of model weights, *i.e.*, $\boldsymbol{\theta} + \mu\mathbf{u}_i$ or $\boldsymbol{\theta} + \mu\mathbf{e}_i$, ③ model inference for calculating finite differences, and ④ other arithmetic operations for gradient estimation. Our results show that RGE takes much longer than CGE in stages ① and ②. This is because, at each model query, RGE needs to generate a directional perturbation vector with the same dimension as $\boldsymbol{\theta}$ and apply this perturbation

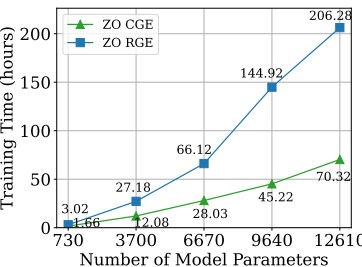

Figure A2: The computation cost of using CGE and RGE to train the simple CNN in Fig. 2 over 50 epochs on CIFAR-10 against parameter size.

to $\boldsymbol{\theta}$. By contrast, CGE can generate the basis vector for 'free' and perturb the model weights using a coordinate-wise indexing operation. **Tab. A1** shows a comparison of RGE and CGE in ①-④. It is also worth noting that a concurrent study (Malladi et al., 2023) argues that language model fine-tuning using RGE only necessitates the query budget $q = 1$. However, it is important to highlight that their approach has certain restrictions, as it relies on having a high-quality pretraining model and a meticulously crafted prompt.

**Tab. A1** provides a comparison of the average computation time per iteration for RGE and CGE-based ZO training across four stages: ① generation of directional vectors, *i.e.*, $\mathbf{u}_i$ or $\mathbf{e}_i$ in (1), ② perturbation of model weights, *i.e.*, $\boldsymbol{\theta} + \mu\mathbf{u}_i$ or $\boldsymbol{\theta} + \mu\mathbf{e}_i$, ③ model inference for calculating finite differences, and ④ other arithmetic operations for gradient estimation. The comparison is conducted for training models of different numbers of parameters ($d$), ranging from 730 to 12610. In stage ①, RGE incurs computation time due to the generation of random Gaussian vectors, whereas CGE is free of this step since it estimates gradients in a coordinate-wise manner. In stage ②, RGE requires more computation time compared to CGE across all parameter sizes. This is because RGE perturbs

Table A1: Average per-iteration computation time comparison (seconds) between RGE and CGE. Here the gradient estimation process is dissected into 4 stages: ① generation of directional vectors (DV), ② model weights perturbation (WP), ③ model inference (MI) for calculating finite differences, and ④ other arithmetic operations (AO) for gradient estimation.

| Parameter # | RGE ($q = d$) / CGE | | | |
|---|---|---|---|---|
| ($d$) | ① DV | ② WP | ③ MI | ④ AO |
| 730 | 0.173 / 0 | 0.152 / 0.026 | 0.202 / 0.21 | 0.149 / 0.0312 |
| 3700 | 1.43 / 0 | 1.2 / 0.0907 | 2.11 / 1.98 | 1.1 / 0.124 |
| 6670 | 3.49 / 0 | 2.93 / 0.159 | 4.84 / 4.53 | 2.68 / 0.225 |
| 9640 | 6.74 / 0 | 5.47 / 0.246 | 8.3 / 8.09 | 4.97 / 0.336 |
| 12610 | 10.2 / 0 | 8.86 / 0.457 | 13.1 / 14.6 | 8.05 / 0.566 |

all model weights in a single query, while CGE only perturbs a single model weight coordinate per query. In stage ③, the computation times for both ZO algorithms increase with the number of parameters, with similar time consumption. In stage ④, RGE continues to require more computation time than CGE for all parameter sizes. This is because RGE needs to perform extra arithmetic operations (such as the sum operation) to aggregate multiple queries, whereas CGE handles multiple queries coordinate-wise, resulting in a more efficient computation.

# D  PERFORMANCE OF MODEL PRUNING VIA ZO-GRASP

We implement ZO-GraSP using RGE with the query number $q = 192$ (smaller than the model size $d$) and compare its performance with random pruning and FO-GraSP. All GraSP variants are performed over randomly initialized model parameters ($\boldsymbol{\theta}$). **Tab. A2** and **Tab. A3** present the pruning performance vs. model pruning ratios in the data-model setup of (CIFAR-10, ResNet-20) and (CIFAR-10, ResNet-18), respectively. Here the pruning performance is evaluated by measuring the testing accuracy of a sparse model on CIFAR-10. To assess the impact of model pruning, we utilize FO SGD as the optimizing method for sparse model training. As we can see, our proposed ZO-GraSP method demonstrates comparable testing accuracy to FO-GraSP across various pruning ratios and significantly outperforms random pruning.

Table A2: Performance comparison of different model pruning methods on (CIFAR-10, ResNet-20).

| Pruning ratio | 10% | 20% | 30% | 40% | 50% | 60% | 70% | 80% | 90% | 95% |
|---|---|---|---|---|---|---|---|---|---|---|
| | | | | | (CIFAR-10, ResNet-20) | | | | | |
| **Random** | $92.71 \pm 0.18$ | $92.63 \pm 0.04$ | $92.40 \pm 0.14$ | $91.94 \pm 0.03$ | $91.77 \pm 0.16$ | $91.41 \pm 0.20$ | $90.57 \pm 0.05$ | $89.35 \pm 0.19$ | $86.55 \pm 0.15$ | $82.51 \pm 0.16$ |
| **FO-GraSP** | $92.64 \pm 0.10$ | $92.44 \pm 0.06$ | $92.34 \pm 0.11$ | $92.35 \pm 0.14$ | $92.07 \pm 0.11$ | $92.00 \pm 0.03$ | $91.63 \pm 0.16$ | $90.75 \pm 0.07$ | $88.55 \pm 0.09$ | $85.48 \pm 0.26$ |
| **ZO-GraSP** | $92.74 \pm 0.07$ | $92.56 \pm 0.10$ | $92.58 \pm 0.20$ | $92.46 \pm 0.05$ | $92.09 \pm 0.10$ | $91.61 \pm 0.13$ | $91.48 \pm 0.23$ | $90.21 \pm 0.16$ | $88.08 \pm 0.09$ | $84.80 \pm 0.10$ |

Table A3: Performance comparison of different model pruning methods on (CIFAR-10, ResNet-18).

| Pruning ratio | 10% | 20% | 30% | 40% | 50% | 60% | 70% | 80% | 90% | 95% |
|---|---|---|---|---|---|---|---|---|---|---|
| | | | | | (CIFAR-10, ResNet-18) | | | | | |
| **Random** | $95.45 \pm 0.06$ | $95.51 \pm 0.11$ | $95.32 \pm 0.15$ | $95.34 \pm 0.10$ | $95.40 \pm 0.20$ | $95.09 \pm 0.13$ | $94.92 \pm 0.14$ | $94.58 \pm 0.06$ | $93.53 \pm 0.10$ | $92.13 \pm 0.26$ |
| **FO-GraSP** | $95.44 \pm 0.07$ | $95.46 \pm 0.30$ | $95.53 \pm 0.06$ | $95.49 \pm 0.05$ | $95.44 \pm 0.08$ | $95.86 \pm 0.09$ | $95.99 \pm 0.17$ | $95.97 \pm 0.05$ | $95.91 \pm 0.03$ | $96.19 \pm 0.10$ |
| **ZO-GraSP** | $95.43 \pm 0.05$ | $95.53 \pm 0.03$ | $95.53 \pm 0.14$ | $95.56 \pm 0.18$ | $95.50 \pm 0.06$ | $95.24 \pm 0.09$ | $95.36 \pm 0.05$ | $95.28 \pm 0.05$ | $94.94 \pm 0.08$ | $94.39 \pm 0.13$ |

# E   ALGORITHM DETAILS

---

**Algorithm 1** ZO-GraSP-oriented-LPR-guided ZO training

---

1: Get $\mathcal{S}_{\text{ZO-GraSP}}$ through ZO-GraSP (3)
2: Obtain layer-wise pruning ratio $\mathcal{S}_{\text{layer}}$ based on $\mathcal{S}_{\text{ZO-GraSP}}$
3: **for** Epoch $t = 0, 1, 2, \ldots, T - 1$ **do**
4:     Randomly generate a sparse coordinate set $\mathcal{S}_t$ according to $\mathcal{S}_{\text{layer}}$
5:     **for** Iterations per epoch **do**
6:         Obtain (Sparse-CGE) $\hat{\nabla}_{\boldsymbol{\theta}} \ell(\boldsymbol{\theta})$ based on $\mathcal{S}_t$
7:         Update model weights: $\boldsymbol{\theta} \leftarrow \boldsymbol{\theta} - \alpha \hat{\nabla}_{\boldsymbol{\theta}} \ell(\boldsymbol{\theta})$
8:     **end for**
9: **end for**

---

**ZO-GraSP-oriented-LPR-guided ZO training**   **Algorithm 1** shows the algorithmic steps to fulfill the ZO-GraSP-oriented-LPR-guided ZO training. At initialization, the algorithm acquires the coordinate set of unpruned model weights $\mathcal{S}_{\text{ZO-GraSP}}$ by applying our proposed ZO-GraSP as shown in (3). Subsequently, it computes the layer-wise pruning ratios, denoted as $\mathcal{S}_{\text{layer}}$, derived from $\mathcal{S}_{\text{ZO-GraSP}}$. The (Sparse-CGE) $\hat{\nabla}_{\boldsymbol{\theta}} \ell(\boldsymbol{\theta})$ is then estimated using $\mathcal{S}_{\text{layer}}$. The model weights are then updated by subtracting the estimated sparse gradient to the current weight $\boldsymbol{\theta}$. This process iteratively refines the weights for optimizing the model's performance.

**Fig. A3** demonstrates the superiority of SR-guided training to another two alternative methods, $\mathcal{M}_1$ built on alternative optimization between ZO-GraSP and sparse training, and $\mathcal{M}_2$ based on pruning before model training. To examine the effectiveness of SR in integrating ZO-GraSP with model training, we compare the pruning performance in the FO training regime (*i.e.*, the resulting sparse model is trained using FO methods). As we can see, SR-guided training consistently outperforms methods $\mathcal{M}_1$ and $\mathcal{M}_2$, with the accuracy improvement becoming more pronounced as the pruning ratio increases. As the pruning ratio increases, the performance gap with the original dense model also widens. However, our method achieves the smallest performance gap compared to the FO baseline.

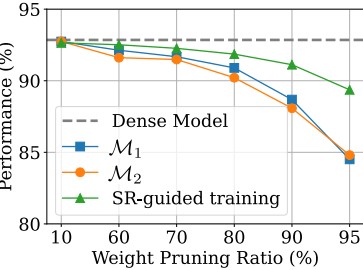

Figure A3:   Performance (testing accuracy) of SR-guided training vs. pruning ratios on (CIFAR-10, ResNet20).

**DeepZero framework.**   We introduce the DeepZero framework, a novel approach designed to train models using ZO optimization. This framework assimilates three key features: the implementation of ZO-GraSP-oriented-LPR-guided ZO training, the reuse of features, and the adoption of parallel computation techniques. These components of the DeepZero framework are expounded in **Algorithm 2**.

---

**Algorithm 2** DeepZero Framework

---

1: **Initialize:** Total epochs $T$, sparsity update interval $K_{\text{sparse}}$, LPRs $\mathcal{R}$ via ZO-GraSP (3)
2: **for** Epoch $t = 0, 1, 2, \ldots, T - 1$ **do**
3:     **if** $t \bmod K_{\text{sparse}} == 0$ **then**                        ▷ Sparsity inducing
4:         Update sparse coordinate index set $\mathcal{S}_t$ according to $\mathcal{R}$
5:     **end if**
6:     **for** Process $i = 1, 2, \ldots, M$ **do**                        ▷ $M$ processes
7:         Based on $\mathcal{S}_t$, parallelized forward evaluations with feature reuse (4)-(5)
8:     **end for**
9:     (Sparse-CGE) estimation using above model evaluations
10:     Model weights updating and synchronization
11: **end for**

---

# F ADDITIONAL EXPERIMENTS OF IMAGE CLASSIFICATION

**Fig. A4** presents the training trajectory comparison between DeepZero and the dense FO training baseline. We observe that despite the marginal performance drop, DeepZero achieves a competitive convergence rate as the FO counterpart. Besides, it is important to highlight that due to the small variance of CGE on a reduced dimension (parameters after pruning), DeepZero achieves competitively stable training as the FO baseline, as indicated by similar standard deviations.

**Fig. A5** presents the performance and the time consumption of DeepZero in various batch size settings. The impact of batch size on model performance has been well-documented in FO training, with larger batch sizes often leading to decreased performance. We sought to examine the effect of batch size selection on ZO training and determine whether increasing the size could reduce training time. As depicted in **Fig. A5**, we observed a marked decline in performance when the batch size exceeded 2048. Additionally, when the batch size was set to 512, it reached a constant time consumption of 60 minutes per epoch, rendering any further reduction in training time infeasible due to the limitations of GPU computational capacity. We stress that reaching this constant time consumption signs the full utilization of each GPU, which is easily achieved by forward parallelization.

**Fig. A6** presents DeepZero's training speed (iterations per hour) on CIFAR-10 and ResNet-20 when using different counts of GPUs. We observe training speed scaling linearly with regard to the GPU counts, which justifies the efficiency of acceleration by forward pass and feature reuse.

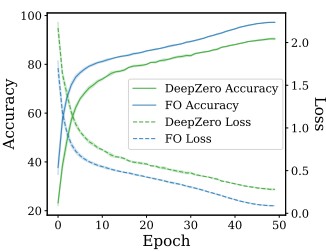 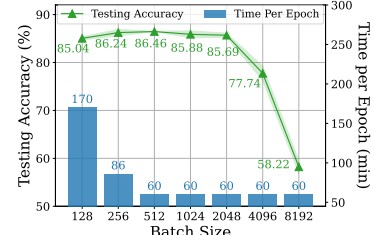 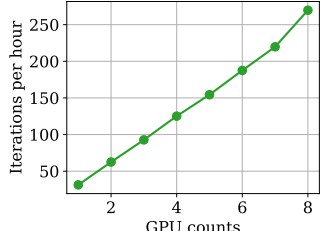

Figure A4: Training trajectory of DeepZero and FO training on (CIFAR-10, ResNet-20). DeepZero adopts Sparse-CGE of 90% sparsity. The mean and standard deviation of 3 independent runs are reported.

Figure A5: Accuracy performance and computation time of DeepZero on (CIFAR-10, ResNet-20) against different batch sizes. Other settings are consistent with Fig. A4.

Figure A6: The training speed of DeepZero on (CIFAR-10, ResNet-20) against GPU counts. Experiments are conducted under NVIDIA TESLA V100 16G GPU resources.

# G ADDITIONAL ILLUSTRATION OF BLACK-BOX DEFENSE AGAINST ADVERSARIAL ATTACKS

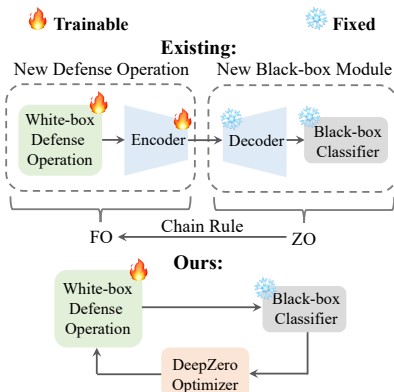

Figure A7: Schematic overview of baseline ZO-AE-DS (Zhang et al., 2022c) and DeepZero-enabled black-box defense.

In **Fig. A7**, we compare our method, DeepZero, with the baseline method ZO-AE-DS (Zhang et al., 2022c). ZO-AE-DS utilizes an autoencoder (AE) to reduce the dimensionality of the ZO gradient estimation by merging the AE's decoder with the black-box classifier and integrating the AE's encoder with the white-box defense operation. However, ZO-AE-DS suffers from poor scalability to high-resolution datasets like ImageNet due to the use of AE, which compromises the fidelity of the image input to the black-box classifier. In contrast, DeepZero directly optimizes the white-box defense operation without the need for an AE, as demonstrated in Fig. A7.

Following (Salman et al., 2020; Zhang et al., 2022c), we model the defense operation as a denoiser given by DnCNN (Zhang et al., 2017). To optimize it for black-box defense, we adopt the 'pretraining + fine-tuning' approach, as detailed in (Zhang et al., 2022c). We first employ the Adam

optimizer (Kingma & Ba, 2014) to pretrain the denoiser over 90 epochs, based on the mean squared error (MSE) loss function. The pre-training stage excludes the optimization with the black-box image classifier. After the pretraining phase, we apply ZO-GraSP to the pretrained denoiser, with the pruning ratio of 95%. Notably, the percentage of unpruned parameters of the denoiser tends to be almost zero for all layers, except the first and the last layers. With this insight in mind, we specify the finetuning strategy as partial finetuning, which targets only the last layer of the denoiser. However, we still need to address the black-box optimization challenge, provided by the black-box classifier $f(\mathbf{x})$. Let $\boldsymbol{\theta}$ represent the denoiser to be finetuned, DeepZero then solves the following black-box optimization problem:

$$\underset{\boldsymbol{\theta}}{\text{minimize}} \quad \mathbb{E}_{\boldsymbol{\delta},\mathbf{x}}[\ell_{\text{CE}}(f(D_{\boldsymbol{\theta}}(\mathbf{x}+\boldsymbol{\delta})), f(\mathbf{x}))], \tag{A2}$$

where $\mathbf{x}$ signifies the input, $\ell_{\text{CE}}$ refers to the Cross-Entropy (CE) loss, and $\boldsymbol{\delta} \in \mathcal{N}(\mathbf{0}, \sigma^2\mathbf{I})$ represents the standard Gaussian noise with variance $\sigma^2$. We apply DeepZero to solve problem (A2) under 20 training epochs, and use a learning rate of $10^{-5}$, which is reduced by a factor of 10 every four epochs.

## H   SIMULATION-COUPLED DL FOR DISCRETIZED PDE ERROR CORRECTION

The feasibility of training a corrective NN through looping interactions with the iterative partial differential equation (PDE) solver, coined 'solver-in-the-loop' (SOL), has been demonstrated in (Um et al., 2020). By leveraging DeepZero, we can enable the use of SOL with non-differentiable, or black-box, simulators Hu et al. (2019); Fang et al. (2022). We name our method **ZO-SOL**. **Fig. A8** presents a comparison with the baseline method NON, given by non-interactive training out of the simulation loop using pre-generated low and high-fidelity simulation data.

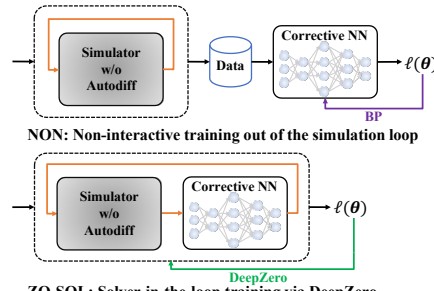

Figure A8: Schematic overview of the baseline approach NON given by training over pre-generated simulation data and ZO-SOL by leveraging DeepZero to address the solver-in-the-loop training challenge.

In our experiments, we consider an unsteady wake flow simulation—a standard benchmark case for fluid dynamics (Um et al., 2020)—to assess the efficacy of our method. The corrective neural network, adopted from (Um et al., 2020), is a sequential convolutional network consisting of two convolutional layers with $5 \times 5$ kernels, totaling 56,898 parameters. The SOL problem can then be formulated as the optimization problem

$$\underset{\boldsymbol{\theta}}{\arg\min} \quad \sum_{t=0}^{T-n}\sum_{i=0}^{n-1} \|\mathcal{P}_s(\tilde{\boldsymbol{s}}_{t+i}) + \mathcal{C}_{\boldsymbol{\theta}}(\mathcal{P}_s(\tilde{\boldsymbol{s}}_{t+i})) - \mathbf{y}_{t+i+1}\|^2 \quad, \tag{A3}$$

where $\mathcal{P}_s$ denotes the PDE solver used for the low-fidelity simulation, $\mathcal{C}_{\boldsymbol{\theta}}$ signifies the corrective neural network parameterized by $\boldsymbol{\theta}$, $\mathbf{y}_{t+i}$ stands for the high-precision simulation state (regarded as the ground truth) at time $t + i$, $T$ is the number of timesteps for the ground truth simulation, $n$ is the number of unrolling steps used in SOL, and $\tilde{\boldsymbol{s}}_{t+i}$ signifies the output from SOL at the $i$th step, namely,

$$\tilde{\boldsymbol{s}}_{t+i} = \underbrace{[(\mathbb{1} + \mathcal{C}_{\boldsymbol{\theta}}) \circ \mathcal{P}s] \circ [(\mathbb{1} + \mathcal{C}_{\boldsymbol{\theta}}) \circ \mathcal{P}s] \circ \cdots \circ [(\mathbb{1} + \mathcal{C}_{\boldsymbol{\theta}}) \circ \mathcal{P}s]}_{i \text{ times}}(\mathbf{y}_t),$$

where $\circ$ is the function composition operation, $\mathbb{1}$ is an identity function and $(\mathbb{1} + \mathcal{C}_{\boldsymbol{\theta}})$ stands for a function expressed as $(\mathbb{1}(\cdot) + \mathcal{C}_{\boldsymbol{\theta}}(\cdot))$. Note that in our experiments, we take $n = 16$ during training, that is we have 16 unrolling steps.

For the unsteady wake flow simulation, we follow the details outlined in (Um et al., 2020) which we include here for completeness. Namely, the unsteady wake flow simulation solves a discretized

version the the incompressible Navier-Stokes equations given by:

$$\frac{\partial u_x}{\partial t} + \boldsymbol{u}\nabla u_x = -\frac{1}{\rho}\nabla p + \nu\nabla \cdot \nabla u_x \tag{A4}$$

$$\frac{\partial u_y}{\partial t} + \boldsymbol{u}\nabla u_y = -\frac{1}{\rho}\nabla p + \nu\nabla \cdot \nabla u_y \tag{A5}$$

$$\nabla \cdot \boldsymbol{u} = 0, \tag{A6}$$

where $t$ is time, $\rho$ is density, $\nu$ is viscosity, $p$ is pressure, $\boldsymbol{u} = (u_x, u_y)^T$ is velocity ($x$ and $y$ directions), and $\nabla\cdot$ denotes the divergence operator. The domain is $\Omega = [0, 1] \times [0, 2]$ with open boundary conditions and an initial state of $\boldsymbol{u} = (0, 1)^T$ along $x \in [0, 1]$ and $y = 0$. Additionally, there is a circular rod of diameter $0.1$ at position $(0.5, 0.5)^T \in \Omega$. The domain $\Omega$ for the low and high fidelity simulations is discretized using a staggered grid of dimensions $[32, 64]$ and $[64, 128]$, respectively. Additionally, we take $T = 500$ timesteps. We utilized the PhiFlow (Holl et al., 2020) simulation code as the solver for our experiments and we based our implementation off of the solver in the loop training process from the code `https://github.com/tum-pbs/Solver-in-the-Loop`.

For simplicity, the loss in (A3) is only expressed for a single simulation with $T$ timesteps. In our experiments, we utilize 6 different simulations in the training dataset each of which have different Reynolds numbers (which affects fluid flow turbulence and can be altered by varying $\nu$ in (A4) and (A5)) in the set $\{97.7, 195.3, 390.6, 781.3, 1562.5, 3125.0\}$, consistent with (Um et al., 2020). Hence, during training the loss function has an additional outer summation to account for the 6 different training simulations. The test set consists of five different simulations computed using Reynolds numbers in the set $\{146.5, 293.0, 585.9, 1171.9, 2343.8\}$, also consistent with (Um et al., 2020).

To solve problem (A3), DeepZero under the Adam optimization framework is used with a learning rate of $10^{-4}$ over ten training epochs.

## I   BROADER IMPACTS AND LIMITATIONS

**Broader Impacts.**   Our proposed ZO learning for deep model training offers a valuable solution for various ML problems that involve interactions with black-box APIs, such as language models as a service, and on-chip learning problems where gradient calculation is infeasible on resource-limited hardware. The applications of DeepZero and its black-box approaches explored in this work can also contribute to advancements in optimization theory and model pruning in other fields. The insights gained from studying DeepZero's scalability, efficiency, and effectiveness can have far-reaching implications beyond the realm of deep learning.

**Limitations.**   One limitation of our approaches is the high number of model queries required, which is inherent to ZO optimization in general. Improving query efficiency is an important area for future research and compute-efficient techniques from (Bartoldson et al., 2023) likely will be helpful. Additionally, the infusion of sparse deep priors in ZO training may not be suitable for non-deep learning models. Therefore, it remains crucial to develop more generic and model-agnostic ZO training methods that can handle large-scale and extreme-scale problems effectively.

