# OpenReview forum: "DeepZero: Scaling Up Zeroth-Order Optimization for Deep Model Training"
_ICLR.cc/2024/Conference — ICLR 2024 poster_

### Official Review · Reviewer_4mR9 · 2023-10-31

**Soundness:** 3 good
**Presentation:** 3 good
**Contribution:** 3 good
**Rating:** 5
**Confidence:** 4

**Summary:**

The paper introduces DeepZero, a framework designed to enhance the scalability of zeroth-order optimization for deep neural network training. The integration of coordinate-wise gradient estimation, sparsity-inducing pruning, feature reuse, and forward parallelization into a unified training pipeline is a novel approach to addressing the scalability challenge of zeroth-order optimization. The paper showcases the effectiveness and efficiency of DeepZero in a wide range of applications, including image classification tasks and practical black-box deep learning scenarios.

**Strengths:**

The paper presents a novel and innovative approach to addressing the scalability challenge of zeroth-order optimization in deep neural network training.
The proposed DeepZero framework is shown to be effective and efficient in a wide range of applications, demonstrating its potential for practical use.
The paper provides a thorough and detailed analysis of the performance of DeepZero, including comparisons to other state-of-the-art approaches.

**Weaknesses:**

1. Lack of clarity and organization in the presentation of the proposed framework and experimental results.
  - The paper could benefit from clearer explanations of the individual components of the DeepZero framework and how they work together.
  - The experimental results could be better organized and presented in a more easily interpretable format.
2. Insufficient comparison to other state-of-the-art approaches.
  - While the paper does provide some comparisons to other approaches, it could benefit from a more thorough and detailed analysis of how DeepZero compares to other state-of-the-art methods.
  - The paper could also benefit from a more detailed discussion of the limitations and potential drawbacks of the proposed approach.
3. Lack of clarity on the broader impact of the paper.
  - The paper briefly mentions the potential impact of DeepZero in other domains, but could benefit from a more detailed discussion of the broader implications of the proposed approach.

**Questions:**

1. Provide clearer explanations of the individual components of the DeepZero framework and how they work together.
2. Organize the experimental results in a more easily interpretable format, such as tables or graphs.
3. Conduct a more thorough and detailed analysis of how DeepZero compares to other state-of-the-art methods.
4. Provide a more detailed discussion of the limitations and potential drawbacks of the proposed approach.
5. Expand on the potential impact of DeepZero in other domains, such as digital twin applications and on-device training.

---

> ### Comment · Reviewer_4mR9 · 2023-11-15
>
> I have read the author rebuttal and made any necessary changes to my review.

---

> > ### Author Response · Authors · 2023-11-23
> > **A kind follow-up**
> >
> > Dear Reviewer 4mR9,
> >
> > We observed your comment, "I have read the author rebuttal and made any necessary changes to my review," dated November 15 (prior to our rebuttal submission). It appears that this comment may have been made unintentionally. On November 16, we submitted our well-prepared responses, and we would like to inquire if you have had an opportunity to review them. If you have found our responses to be informative and helpful, we would greatly appreciate your acknowledgment. Additionally, should you have any further questions or require additional clarifications, please do not hesitate to reach out to us.
> >
> > Thanks,
> >
> > Authors

---

> ### Author Response · Authors · 2023-11-16
> **Response to Reviewer 4mR9 (Part I)**
>
> **Q1: Explainations of individual components, and better organization of experiments**
>
> **A1:** We appreciate your feedback, and we are committed to enhancing the clarity of our paper. However, we respectfully disagree with the assertion that our paper lacks clear explanations of the individual components of the DeepZero framework. Please allow us to elabroate on **the connections of each component and how we presented them in our paper**.
>
> + First, **Sec. 3** aims to elucidate **the rationale behind selecting CGE over RGE as our base ZO gradient estimator**, due to the former’s accuracy and computation efficiency merits. Yet, we still need to tackle the challenge posed by the query complexity of CGE. This inspires us to introduce sparsification in Sec. 4. Thus, CGE is the base gradient estimator component in the DeepZero framework, but requires further improvement on query efficiency (see the last paragraph of Sec. 3).
> + Second, in **Sec. 4,we develop a ZO pruning method, called ZO-GraSP and utilize it to generate layer-wise pruning ratios, and then to construct the query-efficient and sparsity-integrated CGE strategy.** This is eventually used in the DeepZero framework for gradient estimation. We have elucidated the detailed steps of how to integrate sparsity into CGE within the context of ZO learning in Algorithm 1. And the overall DeepZero framework is summarized in Algorithm 2,  which can be found in Appx. E.
> + Third, in **Sec. 5**, we introduce **two implementation techniques to further enhance the efficiency of Zeroth-Order (ZO) training**. The first technique focuses on disentangling intermediate features from weight perturbations, while the second one leverages the finite-difference nature inherent in CGE to facilitate a scalable distributed implementation.
>
> Our experimental results are presented in accordance with the aforementioned components and organizational structure, such as **Fig. 2** and **Fig. 3**. Our experiments are further elaborated upon in **Sec. 6**, where we conduct more comprehensive comparisons with various gradient-free training baselines and apply our approach to a range of distinct black-box machine learning applications.

---

> ### Author Response · Authors · 2023-11-16
> **Response to Reviewer 4mR9 (Part II)**
>
> **Q2: Insufficient comparison to other state-of-the-art approaches.**
>
> **A2:** Thanks for raising this question. However, we respectfully disagree that our work provides an insufficient comparison to other state-of-the-art approaches. By contrast, we have made diligent efforts to compare our approach with a diverse set of gradient-free model training methods to the best of our knowledge.
>
> + First, **it's crucial to highlight that in practical terms, there currently exists no known work capable of harnessing ZO optimization to train a model as complex as ResNet-20 from scratch while achieving comparable accuracy to our results.** As elucidated in **Sec. 1 and 2**, despite the successes of ZO in addressing machine learning challenges, its application has predominantly been confined to relatively small-scale scenarios. When comparing our approach to the gradient-free training baseline introduced in ICLR'23 [R1], it is worth noting that the latter is primarily designed for much smaller-scale applications than ours. It typically targets models like LeNet, especially under conditions of limited data samples and few-shot scenarios. Thus, Fig. 5 in our paper illustrates this comparison within the low-data regime on CIFAR-10.
> + Second, **we have conducted comparisons between our approach and a range of backpropagation-free (bp-free) but computation graph-dependent baselines**. It is worth noting that strictly speaking, these methods may not be suitable for black-box machine learning applications due to their reliance on the computation graph. However, we have included this comparison because these methods demonstrate scalability in training deep neural networks without employing bp. The methods we considered include Align-ada (Boopathy & Fiete, 2022), feedback alignment (FA) (Lillicrap et al., 2014), and direct feedback alignment (DFA) (Nøkland, 2016). We have also provided an explanation for our consideration of Align-ada as a SOTA method within the context of our research, see the last paragraph of Sec. 6.1. This comparison helps to position our approach within the landscape of relevant baselines.
>
> In our exploration of black-box machine learning applications (as detailed in **Sec. 6.2**), we have also conducted comparisons between our approach and task-specific baselines, as evident in **Tab. 2** and **Fig. 6**.
>
> Considering the aforementioned comparisons we have already incorporated into our paper, we do not believe that our work should be characterized as lacking sufficient experimental comparisons. Nevertheless, if Reviewer 4mR9 can identify additional scalable and fair baselines that we should consider for comparison, we are open to conduct additional experiments to further strengthen our research.
>
> [R1] Chiang, Ping-yeh, et al. "Loss landscapes are all you need: Neural network generalization can be explained without the implicit bias of gradient descent." ICLR'23.

---

> ### Author Response · Authors · 2023-11-16
> **Response to Reviewer 4mR9 (Part III)**
>
> **Q4: Detailed limitations and drawbacks.**
>
> **A3:** Thanks for raising this question. **From our perspective, the primary limitation of our work revolves around the challenge of further improving the scalability of ZO training as neural network sizes continue to grow.** In theory, our developed sparsity-induced CGE exhibits a linear relationship with respect to the unpruned weight coordinates, rather than, for example, a sub-linear scaling rate relative to model size. One potential direction for future exploration could involve the development of a hybrid approach that combines CGE with RGE, leveraging RGE's ability to demonstrate a sub-linear relationship between query numbers and parameter size. In our revision, we will provide more detailed discussions about the limitations of our approach. Thanks.
>
>
>
>
>
> **Q5: Expand on the potential impact of DeepZero in other domains, such as digital twin applications and on-device training.**
>
> **A4:** Thanks for raising this question. Please see our further explanations below.
> + In the domain of **digital twins** [R2, R3], physics-informed machine learning plays a crucial role by integrating data from physics-based simulations into machine learning training to enhance the underlying physical systems. For instance, consider our application of "simulation-coupled deep learning for discretized PDE error correction," which can be seen as a form of delta learning [R2] in the context of digital twins. In this approach, a machine learning model learns residuals to refine predictions made by a physics-based model. The final predictions combine the initial model's output with these residuals. This hybrid physics-machine learning approach serves two main purposes: (1) the physics-based model provides baseline predictions based on fundamental principles, which may lack accuracy for individual system units but generalize well across various operating conditions, and (2) the data-driven residuals compensate for missing physics knowledge and unit-to-unit variability. As a result, this approach offers improved generalization compared to pure data-driven machine learning and enhanced prediction accuracy compared to purely physics-based modeling.
> + In the domain of **on-device training**, conducting backpropagation (BP) becomes quite challenging on many edge devices, such as FPGA, ASIC, photonic chips, and embedded microprocessors. These devices often lack the necessary computing and memory resources to support automatic differentiation (AD) libraries. Consequently, ZO optimization has emerged as a potent alternative, as highlighted in related work cited in the Introduction  (Gu et al., 2021b; Gu et al., 2021c). However, adapting ZO optimization for on-device training presents additional complexities, including device constraints and the imperative for power efficiency. Therefore, this remains a research direction that warrants further exploration and investigation at both software and hardware sides.
>
> We will add the above discussions in the revision. Thanks for the insightful comment.
>
> [R2] Thelen, Adam, et al. "A comprehensive review of digital twin—part 1: modeling and twinning enabling technologies." Structural and Multidisciplinary Optimization 65.12 (2022): 354.
>
> [R3] Thelen, Adam, et al. "A comprehensive review of digital twin—part 2: roles of uncertainty quantification and optimization, a battery digital twin, and perspectives." Structural and multidisciplinary optimization 66.1 (2023): 1.

---

> ### Author Response · Authors · 2023-11-21
> **Reminder on follow-up discussion (1 day left before rebuttal ends)**
>
> Dear Reviewer 4mR9,
>
> Thank you very much for the dedicated review of our paper. We are fully aware of the commitment and time your review entails. Your efforts are deeply valued by us.
>
> With only 1 day remaining before the conclusion of the discussion phase, we wish to extend a respectful request for your feedback about our responses. Your insights are of immense importance to us, and we eagerly anticipate your updated evaluation. Should you find our responses informative and useful, we would be grateful for your acknowledgment. Furthermore, if you have any further inquiries or require additional clarifications, please don't hesitate to reach out. We are fully committed to providing additional responses during this crucial discussion phase.
>
> Best regards,
>
> Authors

---

### Official Review · Reviewer_2ocP · 2023-10-31

**Soundness:** 2 fair
**Presentation:** 2 fair
**Contribution:** 2 fair
**Rating:** 3
**Confidence:** 2

**Summary:**

The authors proposed a zeroth-order neural network training method that combines forward-parallelized coordinate-wise difference evaluation and zeroth-order signal-based sparsification at initialization.  They presented experimental results on Resnet-20 trained on CIFAR10.

**Strengths:**

- Zeroth-order optimization holds promise in explaining the biological plausibility of gradient-free learning.  Combined with network pruning, which is another biologically prominent feature, it might be useful for computational neuroscientific work.
- Coordinate-based finite difference estimation coupled with forward parallelization is potentially useful in achieving practicality.

**Weaknesses:**

- Chosen model for experimentation is not complex enough to make a convincing demonstration of the effectiveness of the method.  How does it perform on training transformer architectures of larger size?
- How much sparsification and CGE individually contributed to closing the gap between dense-FO training and sparse-ZO training is not systematically studied.  Independent ablation studies.

**Questions:**

See above.

---

> ### Author Response · Authors · 2023-11-16
> **Response to Reviewer 2ocP**
>
> **W1: The model is not complex enough.**
>
> **A1:** Thank you for raising this question. However, we respectfully disagree that the model considered for ZO training is not complex enough.
>
> **While ResNet-20 may not be considered complex in the FO regime, training a model as seemingly modest as ResNet-20 in the ZO learning field is a notably non-trivial endeavor.** In fact, to the best of our knowledge, there is no existing work that leverages ZO optimization, which only relies on function queries, to train a model like ResNet-20 from scratch while achieving comparable accuracy to our results. As explained in Sec. 1 and 2, despite the successes of ZO in tackling machine learning problems, its application has historically been confined to relatively small-scale scenarios. When comparing our approach to the gradient-free training baseline presented in ICLR'23 [R1], **the latter is primarily constrained to small-scale applications, such as models like LeNet under conditions of low-sample and few-shot scenarios.** Note that LeNet's parameter count stands at approximately 60,000, significantly less than the 270,000 parameters in ResNet-20 that our method utilizes. In light of the above discussion, our work provides an important step in upscaling ZO optimization for DL applications, establishing it as the current SOTA in addressing the challenges posed by black-box neural network training, as outlined in **Sec. 6.2**.
>
> While the ultimate aspiration of the ZO learning field is to scale ZO techniques to any complex models, it is important to acknowledge that practical advancements in this direction have yet to fully materialize. We will incorporate this discussion into conclusion and future work. Thank you very much for your valuable insights and feedback.
>
>
> **W2: How much sparsification and CGE individually contributed to closing the gap between dense-FO training and sparse-ZO training?**
>
> **A2:** Thanks for raising this question. To clarify, we have made several efforts to peer into the contributions of sparsification and CGE to reducing the performance gap with dense-FO training.
>
> + First, regardless of the sparsification, **we have conducted a performance comparison between ZO-CGE and dense FO training** using a simple CNN with varying numbers of parameters, as depicted in **Fig. 2**. In this scenario, we were able to accommodate the query cost associated with the full CGE computation. As **Fig. 2** illustrates, CGE indeed exhibits a significantly smaller performance gap than RGE when compared to FO training. Additionally, as demonstrated in **Appx. C**, CGE also showcases computational efficiency advantages relative to RGE. Hence, based on the above considerations, we have chosen CGE as our base ZO gradient estimator. Yet, we still need to tackle the challenge posed by its query complexity. This inspires us to introduce sparsification in **Sec. 4**.
> + **The contribution of sparsification has been investigated both in theory and through empirical experiments.** As detailed in **Appx. 1**, sparsity presents a tradeoff: as the sparsity ratio, denoted as $p$, increases, the query complexity decreases, but the potential for convergence errors may also rise. In our empirical investigations, **Fig. 4** illustrates the performance of our proposed ZO training approach, DeepZero, alongside dense FO training and various sparse variants of FO training at different sparsity ratios. When we compare DeepZero with "FO training with Sparse Weight" in **Fig. 4**, a notable observation is that our performance is comparable to and could even potentially surpass that of the latter. This is attributed to DeepZero's better sparsity-inducing mechanism during training, which outperforms the approach of training a sparse network with a fixed sparsity pattern.
>
> [R1] Chiang, Ping-yeh, et al. "Loss landscapes are all you need: Neural network generalization can be explained without the implicit bias of gradient descent." ICLR'23.

---

> ### Author Response · Authors · 2023-11-21
> **Reminder on follow-up discussion (1 day left before rebuttal ends)**
>
> Dear Reviewer 2ocP,
>
> Thank you very much for the dedicated review of our paper. We are fully aware of the commitment and time your review entails. Your efforts are deeply valued by us.
>
> With only 1 day remaining before the conclusion of the discussion phase, we wish to extend a respectful request for your feedback about our responses. Your insights are of immense importance to us, and we eagerly anticipate your updated evaluation. Should you find our responses informative and useful, we would be grateful for your acknowledgment. Furthermore, if you have any further inquiries or require additional clarifications, please don't hesitate to reach out. We are fully committed to providing additional responses during this crucial discussion phase.
>
> Best regards,
>
> Authors

---

### Official Review · Reviewer_r5je · 2023-11-01

**Soundness:** 2 fair
**Presentation:** 3 good
**Contribution:** 3 good
**Rating:** 6
**Confidence:** 3

**Summary:**

This paper proposes DeepZero, a framework that aims for scalable training from scratch with ZO. The main contributions are in 2-fold:

(1) the authors demonstrate that coordinate-wise gradient estimation (CGE) would outperform randomized gradient estimation (RGE) under the same number of model queries, and CGE is more computationally-efficient and performs better under the same number of model queries than RGE. In addition, the authors propose to reuse the layerwise features before the perturbed parameter's model layer, and parallelize the forward computation in multiple processes.

(2) the authors propose the ZO-GraSP algorithm that uses ZO to estimate Hessian-gradient product for each parameter for the GraSP algorithm, and select a subset of parameters to perform the ZO update with CGE.

The authors perform experiments on ResNet-20 (and 8-layer CNN) on CIFAR10, black-box defense against adversarial attacks with DnCNN and ResNet-50 on ImageNet-10, and corrective NN with iterative PDE solver for simulating unsteady wake flow. In the image classification task, the authors also perform comprehensive scaling experiments with different sparsity ratio, parameter size, dataset size, batch size, and number of GPUs, and demonstrate the performance and scalability of deepzero framework.

**Strengths:**

The motivation of this paper is good as the problem is interesting for the wider optimization community: traditional ZO methods do not focus on training a large NN from scratch due to a dependency over $d$.

The method of reusing intermediate features is effective in accelerating CGE. This method is also tailored to CGE instead of RGE as RGE would perturb all layers, and therefore change the intermediate features of all layers per each query.

Choosing a sparse set of indices to perform ZO update would be computationally-efficient to the ZO-CGE method. I also believe the DeepZero could be more general than only the ZO-GraSP method. In this case, DeepZero could be a general framework for training deep models with ZO from scratch.

The scaling experiments in the image-classification task are quite comprehensive.

**Weaknesses:**

- Although the authors make a good comparison of CGE vs. RGE with the same number of queries in Figure 2 and Table A1 and argue the computational efficiency of CGE when $q = d$, the authors do not propose a variant to CGE that would require the number of model queries sub-linear to the parameter size (the linear relationship still holds even we have sparse subset of parameters and parallelization). This issue could be fatal as if we have a model with more than 1 million parameters, DeepZero will still take significant wall-clock time to pretrain even under 90% sparsity (in fact, it already takes 28 hrs with 12k parameters in Figure 3).

- An ablation study missing in this paper is to show the performance of RGE versus the number of queries. We should expect such curve to plateau quickly, and still below the performance of CGE across multiple tasks to show the competitiveness of CGE. Ideally, this comparison should be done across models and tasks, as the statement on the advantage of CGE over RGE when $q$ = $d$ is quite strong and we will need a strong evidence to justify this argument.

- The authors claim the idea of "forward parallelization" for ZO-CGE as the novelty of this paper. Prior research (e.g. the last sentence of section 3 in Ruan et al., 2019; the second paragraph of section 1.2 in Cai et al. 2021) have indicated that the gradient estimator of ZO (both CGE and RGE) is parallelizable. However, I did not find any prior literature indicating the intermediate feature reuse of CGE, and I would accept this novelty claim at this moment.


References:

- Ruan, Yangjun, et al. "Learning to Learn by Zeroth-Order Oracle." International Conference on Learning Representations. 2019.

- Cai, HanQin, et al. "A zeroth-order block coordinate descent algorithm for huge-scale black-box optimization." International Conference on Machine Learning. PMLR, 2021.

**Questions:**

- Is it possible to use other pruning methods to select sparse subset with ZO than only GraSP method?

Balancing the strength and weakness of this paper, I am inclined to give a weak accept score. But I am happy to raise my score if the above concerns are addressed.

---

> ### Author Response · Authors · 2023-11-16
> **Response to Reviewer r5je (Part I)**
>
> **W1: The number of model queries is not sublinear to the parameter size.**
>
> **A1:** We appreciate the observation that our proposed ZO training method necessitates a linear increase in the number of queries concerning the **unpruned** weight coordinates, rather than exhibiting a sublinear relationship with parameter size.
>
> Indeed, the sub-linear scaling advantage of CGE concerning model size becomes particularly appealing when dealing with significantly larger model parameters. **While achieving this ultimate goal is the aspiration of the ZO learning field, practical advancements have not yet reached that level.**
>
> In theory, the development of CGE with a sub-linear scaling rate relative to model size remains an open question. Importantly, in practical terms, there is currently no existing work capable of leveraging ZO optimization to train a model like ResNet-20 from scratch while achieving comparable accuracy to our results. As discussed in **Sec. 1 and 2**, despite the successes of ZO in addressing machine learning problems, its application has been primarily limited to relatively small-scale scenarios. When comparing our approach to the gradient-free training baseline introduced in ICLR'23 [R1], it becomes evident that the latter is primarily tailored to much smaller-scale applications than ours such as models like LeNet, particularly under conditions of limited data samples and few-shot scenarios.
>
> In light of the above discussion, our work takes an important step to scale up ZO optimization. This positions our approach as the current SOTA in effectively addressing the challenges associated with black-box neural network training, as elaborated upon in **Sec. 6.2**.
>
> Building upon the insightful comment provided by the reviewer, we acknowledge that the quest to find strategies to decouple the linear relationship between the number of queries and parameter size in CGE remains a crucial avenue for future research. One potential direction worth exploring could involve the development of a hybrid approach that combines CGE with Randomized Gradient Estimation (RGE), leveraging RGE's capability to exhibit a sub-linear relationship between query numbers and parameter size. Thanks for this insightful comment.
>
>
>
> **W2: Missing ablation study on the query number of RGE.**
>
> **A2:** Thank you for bringing this to our attention. In **Tab. R1**, we have incorporated an additional ablation study focused on the performance of RGE vs. the query number $q$. As we can see, a larger value of $q$ improves the optimization accuracy. This is not surprising as shown in [R2], it is established that increasing $q$ leads to a reduction in the convergence error on the order of $\sqrt{1/{q}}$.
>
> Given the above, we would like to respectfully re-emphasize the rationale behind comparing CGE with RGE specifically at the setting where $q = d$. As shown in **Sec. 3**, the variance of RGE diminishes and converges toward that of CGE when $q = d$. Hence, assuming we employ a sufficiently large number of random samples, we are interested in exploring the comparative performance of RGE against CGE. This investigation is aimed at helping us determine which ZO gradient estimator is better at preserving model accuracy.
>
> **Table R1:** Ablation study on query number (q) of RGE when training the  Multi-layer CNN model with 6,670 parameters on CIFAR-10, with the same setup as **Fig. 2**.
> | Query Number ($q$) | 6,670 | 4,000 | 2,000 | 1,000 | 500 | 250 | 100 | 50 | 25 | 10 | 5 | 1 |
> |------------------|-------|-------|-------|-------|-----|-----|-----|----|----|----|---|---|
> | Acc (%)          | 65.31 | 55.13 | 55.11 | 53.15 | 52.10| 48.89|46.61|46.58|42.75|38.03|35.14|25.68|
>
> [R1] Chiang, Ping-yeh, et al. "Loss landscapes are all you need: Neural network generalization can be explained without the implicit bias of gradient descent." ICLR'23.
>
> [R2] Liu, Sijia, et al. "Zeroth-order online alternating direction method of multipliers: Convergence analysis and applications." International Conference on Artificial Intelligence and Statistics. PMLR, 2018.

---

> ### Author Response · Authors · 2023-11-16
> **Response to Reviewer r5je (Part II)**
>
> **W3: Novelty of forward parallelization.**
>
> **A3:** Thank you very much for pointing out these related work. Specifically, the work [R3] and [R4] highlights the inherently parallelizable nature of ZO  optimization. However, it is noteworthy that [R3] primarily discusses this aspect in the context of RGE. In contrast, CGE presents a more favorable option for parallelization due to its coordinate-wise fashion, leading to minimal parameter changes per query. While [R4] acknowledges that parallelization can alleviate computational overhead, it falls short of providing detailed methodologies or practical implementations for achieving this. We have already referenced [R3] in the introduction and related work sections of our paper (Cai et al., 2021), and appreciating your suggestion, will include [R4] as well. Thank you for highlighting these crucial citations.
>
> **Q1: Can other pruning methods be used for the proposed ZO training method?**
>
> **A3:** Indeed, there are various pruning methods, roughly categorized as pruning-at-initialization (PAT), pruning-during-training, and pruning-post-training. **In ZO learning, we require a query-efficient pruning approach without access to a pre-trained model (as we aim to solve the DNN training problem).** Hence, the PAT technique fits, with GraSP being a standout choice. GraSP offers an analytical expression based on first- and second-order derivatives (Eq. 2), and thus can be efficiently approximated by ZO gradient estimates (Eq. 3). **While other PAT methods like SNIP and SynFlow are viable, GraSP has been shown with better pruning quality than SNIP, and SynFlow integration with ZO optimization is much more complex than ZO-GraSP.**
>
>
>
> [R3] Cai, HanQin, et al. ”A zeroth-order block coordinate descent algorithm for huge-scale black-box optimization.” International Conference on Machine Learning. PMLR, 2021.
>
> [R4] Ruan, Yangjun, et al. ”Learning to Learn by Zeroth-Order Oracle.”International Conference on Learning Representations. 2019.

---

> ### Author Response · Authors · 2023-11-22
> **Reminder on follow-up discussion (1 day left before rebuttal ends)**
>
> Dear Reviewer r5je,
>
> Thank you very much for the dedicated review of our paper. We are fully aware of the commitment and time your review entails. Your efforts are deeply valued by us.
>
> With only 1 day remaining before the conclusion of the discussion phase, we wish to extend a respectful request for your feedback about our responses. Your insights are of immense importance to us, and we eagerly anticipate your updated evaluation. Should you find our responses informative and useful, we would be grateful for your acknowledgment. Furthermore, if you have any further inquiries or require additional clarifications, please don't hesitate to reach out. We are fully committed to providing additional responses during this crucial discussion phase.
>
> Best regards,
>
> Authors

---

> > ### Comment · Reviewer_r5je · 2023-11-23
> > **Response to authors**
> >
> > Thanks for your insightful response!
> >
> > **A1**
> > I agree that this would be an interesting future direction, but I am concerned that this could be fatal to the computational advantage of CGE over RGE: CGE inherently requires linear scaling when we scale up the number of model parameters (even with pruning), while the query number of RGE is typically independent of the number of model parameters.
> >
> > **A2**
> > Thank you for the ablation experiment! My concern is that the RGE does not converge and still shows a trend to further improve when we scale up $q$. This will weaken the computational advantage of CGE over RGE when we scale up $d$. I also agree with Review 2ocP's first point that finding better performance on one model and one task is not sufficient.
> >
> > **A3**
> > I agree that R4 does not provide a practical implementation, but I don't consider such practical implementation as a solid novelty of this paper. I would recommend including a short discussion on Section 5 and potentially modifying the novelty claim to cause less confusion.
> >
> > **A4**
> > Thank you for your insightful response! It would be better to discuss alternatives to GraSP in the main paper to better inform future research.
> >
> > Overall, I am still inclined to maintain my score, as I believe that the current experiments presented in the paper and during the rebuttal period are not adequate in justifying that the computational advantage of CGE over RGE is universal. This is a fairly strong claim, and it would be nice to see either comprehensive experiments across models and tasks or a theoretical analysis of CGE's computational advantages.

---

> ### Author Response · Authors · 2023-11-23
> **Thank you and further response**
>
> Thank you very much for providing additional feedback. Before the rebuttal deadline, we would like to clarify the following points further.
>
> **Further Response to A1:** We agree that when compared to CGE, the query count of RGE is typically independent of the number of model parameters. Nevertheless, our experiments have consistently demonstrated a noticeable drop in accuracy when using RGE in comparison to CGE. This discrepancy remains even when we increase the query count of RGE to match the model size 'd' (this serves the motivation behind Figure 2). Therefore, when assessing accuracy, our preference lies in employing CGE as our base estimator. However, we acknowledge the valid concern regarding CGE's query complexity, which serves as the motivation behind the sparsity-induced proposal in Section 4.
>
>
> **Further Response to A2:** Your concern about RGE not converging and showing potential for further improvement as 'd' scales up is precisely why we compared RGE with CGE at 'q = d' in Figure 2. Even in this scenario, the accuracy loss associated with RGE during model training remains evident. We do not believe that this diminishes the computational advantage of CGE over RGE. From an accuracy perspective, RGE indeed demands a large 'q' to ensure the quality of gradient estimation, as the variance relies on 'q'. In such cases, we need to generate a random direction vector multiple times. Inherent to this process, generating a random vector of the model's size introduces a higher computational cost compared to the deterministic coordinate-wise perturbation vector used in CGE (see Tab. A1).
>
>
> We acknowledge that achieving better performance on a "single" model and a "single" task may not be sufficient. However, to the best of our knowledge, this is the first work to investigate the effectiveness of ZO optimization for deep learning training from scratch. Consequently, even if our experiments do not encompass "all" scenarios, they provide valuable new insights based on the results obtained so far, as we emphasized in our response to A1. On the other hand, as we responded to Reviewer 2ocP in [A1](https://openreview.net/forum?id=qBWhjsNPEY&noteId=MHXanXFQFU), we have considered “multiple” models and “multiple” tasks, e.g., ResNet-20 (Fig. 4 and 5), CNN of different widths (Table 1), black-box defense task (Table 2), and simulation-coupled DL task (Figure 6).
>
> **Further Response to A3:** Thank you for your valuable suggestion. In response, we will incorporate a brief discussion to refine our novelty claim and provide a more precise statement of our contributions.
>
> **Further Response to A4:** We appreciate your insightful suggestion. We will certainly include a discussion explaining why we chose GraSP and explore alternatives for future research in the revised manuscript.
>
> Last but not least, it's important to clarify that we did **not** claim that the computational advantage of CGE over RGE is **universal**. Rather, our intention was to shed light on the possible computational benefits of CGE over RGE, **specifically in the context of deep model training from scratch**. Given the limited research on scaling up ZO optimization for deep model training, we view our work as a solid step forward in this domain. Furthermore, we have made diligent efforts to ensure comprehensive experimentation (we sincerely hope the reviewer agrees with that), including comparisons with a range of baselines (such as the ZO pattern search baseline for DL training in ICLR'23 [R1] and the computation graph-dependent backpropagation-free training) and the assessment of our approach in two black-box machine learning applications.
>
> [R1 Chiang, Ping-yeh, et al. "Loss landscapes are all you need: Neural network generalization can be explained without the implicit bias of gradient descent." ICLR'23.

---

> > ### Comment · Reviewer_r5je · 2023-11-23
> > **Thank you for the additional clarification**
> >
> > Thank you for the detailed responses! I am overall positive about the pioneering contribution this paper makes, and I fully appreciate the meticulous explanations provided by the authors during the rebuttal period. I am still inclined to keep my score unchanged at this moment, and I will discuss it with other reviewers in the upcoming discussion period.

---

### Official Review · Reviewer_pJQr · 2023-11-01

**Soundness:** 3 good
**Presentation:** 3 good
**Contribution:** 3 good
**Rating:** 8
**Confidence:** 4

**Summary:**

This paper proposed to improve Zero-Order optimization in deep nerual network learning. It first points out that Coordinate-wise Gradient Estimation (CGE) is better than Randomized Vector-wise Gradient (RGE). Then due to the query complexity of CGE is related to the parameter dimension, it propsed to introduce sparsity during ZO training. It used a combination of ZO and GraSP to measure the importance of parameters for pruning.

**Strengths:**

1. Connection between motivation and corresponding method is clear and sound: the fessibility of a better ZO algorithm leads to a sparse solution.
2. Combination of GraSP and ZO is interesting.

**Weaknesses:**

1. Sparsity acts as a tradeoff between performance and fessibility: to get faster training speed require more sparse neural network. A more ideal method is to update portion of parameters at a time while keeping model dense.
2. Author should consider combination of ZO and other sparsity-inducing method, such as OBD [1] and OBS [2] (approximated by ZO). It would add novelty if the proposed method is disentangled with specific pruning method, or it can be provided that ZO take effect only with GraSP,

[1] https://proceedings.neurips.cc/paper/1989/hash/6c9882bbac1c7093bd25041881277658-Abstract.html
[2] https://arxiv.org/abs/1705.07565

**Questions:**

1. It seems impossible that ResNet-20 with 99% reach a 70% accuracy.

**Details Of Ethics Concerns:**

N.A.

---

> ### Author Response · Authors · 2023-11-16
> **Response to Reviewer pJQr**
>
> **W1:A more ideal method is to update portions of parameters at a time while keeping the model dense.**
>
> **A1:** Thank you for your suggestion. To clarify, our approach does **not** involve first pruning a large model to a small one and then applying ZO optimization to the pruned model for training. Instead, we use ZO-GraSP (Sec. 4) to determine the sparsity pattern before training and use it to determine the layer-wise pruning ratios (LPRs) for ZO training. As described in the last paragraph of **Sec. 4** and  **Algorithm 1 (Appx. E)**, we generate a **dynamic coordinate-wise sparse pattern based on LPRs** and utilize it to construct Sparse-CGE. This dynamic sparsity is applied to ZO gradient estimation for **dense model training**. As illustrated in **Fig. 4**, aligning with the reviewer's comment, incorporating gradient sparsity while maintaining a dense model configuration (corresponding “sparse gradients” in **Fig. 4**)  yields better results compared to training a pruned model directly with a fixed weight sparse pattern (corresponding to “sparse weights”).
>
> We also agree that there exists a tradeoff between sparsity and accuracy. In addition to empirical results, this was also studied in terms of convergence rate in **Appx. A**.
>
>
>
> **W2: Weakness in the lack of study on other sparsity-inducing methods.**
>
> **A2:** We appreciate the suggestion for exploring other sparsity-inducing methods. However, we would like to clarify that ZO-GraSP, which is the ZO extension of GraSP, is a "pruning-at-initialization" method quite suited for ZO optimization. It is executed only once, prior to training, at a random initialization, and is used to determine the layer-wise pruning ratios for generating the dynamic gradient sparse pattern in Sparse-CGE and for ZO DNN training from scratch. This has been explained in **Sec. 4**. In the ZO context, we consider the DNN as a black box with no access to backpropagation when training from scratch. **This means pruning-after-training methods like OBD and OBS are not applicable as these methods rely on trained models.** By contrast, ZO-GraSP leverages first-order or second-order derivative estimates  to acquire the weight sparse pattern prior to training.
>
>
>
> **Q1: Impossible 70% performance of ResNet-20 with 99% sparsity.**
>
> **A3:** Thank you for bringing up this question. To clarify, the 99% sparsity in our method pertains to **gradient sparsity**, as explained in the **response A1**. **This sparsity level governs the proportion of parameters within a dense model** that undergo updating per iteration. Therefore, we are able to achieve a 70% accuracy with this configuration. Moreover, as shown in **Fig. 4**, the 99\%-pruned ResNet-20 (with the static sparsity pattern) can indeed achieve a decent accuracy, e.g., 68.22% for FO training with sparse weights.

---

> ### Author Response · Authors · 2023-11-22
> **Reminder on follow-up discussion (1 day left before rebuttal ends)**
>
> Dear Reviewer  pJQr,
>
> Thank you very much for the dedicated review of our paper. We are fully aware of the commitment and time your review entails. Your efforts are deeply valued by us.
>
> With only 1 day remaining before the conclusion of the discussion phase, we wish to extend a respectful request for your feedback about our responses. Your insights are of immense importance to us, and we eagerly anticipate your updated evaluation. Should you find our responses informative and useful, we would be grateful for your acknowledgment. Furthermore, if you have any further inquiries or require additional clarifications, please don't hesitate to reach out. We are fully committed to providing additional responses during this crucial discussion phase.
>
> Best regards,
>
> Authors

---

> ### Comment · Reviewer_pJQr · 2023-11-23
> **Response to Author**
>
> Thanks for your detailed response.
>
> For W1, my point is that: Since the coordinate-wise sparse pattern is dynamic, it is very natural to incorporate gradual sparse training (gradually increase sparsity as training continues). It is a suggestion for improvement.
>
> For W2, it is indeed that pruning-after-training may not suite the proposed method. Author may try other pruning-at-initialization methods such as [1].
>
> For Q1, Sparsity of gradient make senses.
>
> Overall, I am satisfied with the response and the origin submission. It is an interesting attempt to combine ZO and sparsity. I raise my score to 8.
>
> [1] Picking Winning Tickets Before Training by Preserving Gradient Flow

---

> > ### Author Response · Authors · 2023-11-23
> > **Thank you for the highly positive feedback and the valuable suggestion!**
> >
> > Dear Reviewer pJQr,
> >
> > We would like to express our gratitude for your meticulous review and the valuable feedback provided during the rebuttal phase. We are delighted to hear that you are satisfied with our response and we greatly appreciate your recognition of our contributions.
> >
> > With regards to W1, we are thankful for your insightful suggestion for improvement. We will certainly incorporate this point into our future work.
> >
> > Regarding W2, we appreciate your reference to the excellent work in GraSP [1]. It’s worth noting that GraSP was indeed considered and extended to operate in the ZO regime, allowing us to generate the sparse mask.
> >
> > Once again, thank you very much for the fruitful discussion.
> >
> > Authors

---

> > > ### Comment · Reviewer_pJQr · 2023-11-23
> > > **Reference Problem**
> > >
> > > Sorry for citing GraSP in my last response. Anyway my point is to compare the proposed method with other pruning-at-initialization (PAI) method. Overall GraSP is merely one of PAI method, it will add more novelty by incoporating more PAI methods in the framework of sparsity-introduced ZO.
> > >
> > > Author may refer to the following for more comparison.
> > >
> > > [1] Sanity-Checking Pruning Methods: Random Tickets can Win the Jackpot
> > > [2] Pruning Neural Networks at Initialization: Why are We Missing the Mark?
> > > [3] Pruning neural networks without any data by iteratively conserving synaptic flow

---

> > > > ### Author Response · Authors · 2023-11-23
> > > > **Thank you for the additional clarification!**
> > > >
> > > > Dear Reviewer pJQr,
> > > >
> > > > Thank you for the additional clarification. We got your point very clearly. In our revision, we will include a discussion concerning the selection of the PAI methods and will explore the possibility of expanding our experiments in this direction.
> > > >
> > > > We genuinely appreciate your valuable suggestion.
> > > >
> > > > Authors,

---

### Official Review · Reviewer_X34v · 2023-11-01

**Soundness:** 3 good
**Presentation:** 2 fair
**Contribution:** 3 good
**Rating:** 8
**Confidence:** 3

**Summary:**

This paper shows that zeroth-order optimization methods (where gradients are calculated using finite differences) can be applied to deep neural networks. They prune the network at initialization using GraSP but use finite differences to compute the Hessian. They only sparsify the gradients and not the weights so that they can train a dense model while reducing the training cost. They apply their method to non-differentiable applications such as black-box defense against adversarial attacks, and simulation-coupled DL for discretized PDE error correction.

**Strengths:**

Zeroth-order optimization methods can be applied to non-differentiable problems, and are a powerful tool for such applications that would otherwise untrainable. There is increasing interest in incorporating PDE-based simulations into deep neural networks, and zeroth-order optimization is an effective tool for training such non-differentiable networks. The description of their approach is clear, and the paper is easy to follow.

**Weaknesses:**

The experiments on differentiable networks like ResNet-20 is distracting and seems unnecessary. If CGE is used with a small enough mu, the gradient should match that of backprop, so it is obvious that the accuracy will match that of first order methods as shown in Figure 2. If the goal of this comparison is to show the effect of sparsification and approximation errors in the finite difference, the authors should perform a thorough ablation study for the values of mu, q, and the sparsity ratio. Otherwise, I recommend removing the experiments on differentiable networks, since the aim of this paper is not to show how zeroth-order methods compare to first-order methods for differentiable problems. Having this comparison at the beginning of the experiments section may lead the readers to think about this comparison and get distracted from the true message of this paper.

**Questions:**

I don't quite understand the rationale behind the comparison of CGE vs. RGE for q=d. Random sampling is a compromise you make when an exhaustive search of the whole space (q=d) is prohibitive. It is obvious that CGE would be better if you can afford to sample the whole space with orthogonal directions (e_i). It is also obvious why RGE is less efficient when q=d. Since the random vectors (u_i) are not orthogonal, the samples are not independent. The real question is how you can increase the accuracy   of estimating the gradient when q << d.

In the experiments the smoothing function mu is set to 5e-3, which seems fairly large. Since the gradient is exact at the limit of mu -> 0, it would seem like a smaller value would be better. How was this value chosen, and what is the motivation for not using a smaller value?

If I understand correctly, both the “black-box defense against adversarial attacks” and “simulation-coupled DL for discretized PDE error correction” involve deep neural networks where part of the forward function is non-differentiable. If so, would it be possible to use finite difference just for the part that is non-differentiable, while using automatic differentiation for the modules that are differentiable? Theoretically, you just need to compute one of the chained Jacobians during backprop for the non-differentiable part using finite difference. I know such an implementation would be practically very painful, but if you could limit the number of parameters that you need to differentiate with finite difference it would result in a much larger reduction than pruning the whole network with GraSP.

---

> ### Author Response · Authors · 2023-11-16
> **Response to Reviewer X34v (Part I)**
>
> **W1: Experiments on differential ResNet-20 are unnecessary.**
>
> **A1:** Thanks for this insightful comment. The inclusion of experiments involving differentiable networks like ResNet-20 serves **dual purposes** in our study.
>
> + First, while **the primary objective of our paper is not to directly compare zeroth-order methods with first-order methods for differentiable problems, conducting first-order (FO) training serves as a valuable reference point or performance "upper bound" for evaluating zeroth-order (ZO) training outcomes.** Furthermore, since we introduce the concept of "sparsity" into ZO training, the comparison with FO training also provides us with valuable insights into the role that sparsity plays in the optimization process. For instance, in **Fig. 4**, when both ZO and FO training are subjected to the same sparsity configuration, it is expected that FO optimization would outperform ZO optimization, as demonstrated by the performance comparison between DeepZero and FO training with Sparse Gradient. **However, a particularly interesting observation arises when we compare DeepZero with FO training utilizing Sparse Weight.** In this scenario, we find that our approach is comparable to and exhibits the potential to outperform FO training. This can be attributed to DeepZero's more effective sparsity-inducing mechanism during training, outperforming the conventional approach of training a sparse network with a fixed sparsity pattern.
> + Second, **the experiments on ResNet-20 demonstrate the scalability merit of our method compared to other query-based gradient-free methods and the accuracy merit compared to computation graph-dependent BP-free methods.** In the context of ZO optimization, training ResNet-20 within the ZO optimization framework is a notably non-trivial endeavor. In fact, to the best of our knowledge, there is no existing work that leverages ZO optimization, which only  relies on function queries, to train a model like ResNet-20 from scratch while achieving comparable accuracy to our results. When comparing our approach to the ZO training baseline presented in ICLR'23 [R1], the latter is primarily constrained to small-scale applications, such as models like LeNet under conditions of low-sample and few-shot scenarios.  Furthermore, in the context of experiments involving differentiable networks, we have a high degree of confidence in the superior accuracy achieved by our method. It surpasses even BP-free baselines that rely on computational graphs (as depicted in **Fig. 1D** and outlined in **Tab. 1**).
>
> Based on the above discussions, we intend to keep the experiments on training differential neural networks.
>
> **W2: Missing ablation study on smoothing parameter (mu), sparsity ratio, and q.**
>
> **A2:** Thank you for mentioning this aspect.
> + **We acknowledge the critical importance of the hyperparameter $\mu$ in our study and have added an ablation study on $\mu$ to ascertain its choice as shown in Tab. R1  below.** These results  reveal that a smaller value of $\mu$ does not necessarily equate to better performance. This is also a known issue in ZO optimization. Although a smaller $\mu$ can reduce the gradient estimation variance in “theory”, in “practice” an excessively small mu may result in the function difference being overshadowed by numerical error, thereby failing to accurately reflect the function differential (see [R2]).  As evident in **Tab. R1**, the selection of a smoothing parameter $\mu = 5e-3$ delivers the highest testing accuracy when compared to other available options. It's noteworthy that both extremely large and small values of $\mu$ can have detrimental effects on performance. Specifically, when $\mu$ is set to $5e-7$, the ZO method fails to converge.
> + Additionally, **our submission also  included an ablation study on the sparsity level, as illustrated in Fig. 4**. Given the sparsity ratio, the number of queries, q, becomes the product of the parameter dimension and the sparsity ratio. Thus, we only demonstrate the performance vs. the sparsity ratio.
>
> **Table R1:** Ablation study on smoothing parameter (mu) of CGE on Multi-layer CNN model with 9,640 parameters on CIFAR-10, where 5e-3 is the default smoothing parameter value.
> | Smoothing Parameter ($\mu$) | 5e-1 | 5e-2 | 5e-3 | 5e-4 | 5e-5 | 5e-6 | 5e-7 |
> |--------------------------|------|------|------|------|------|------|------|
> | Acc (%)                  | 32.31| 57.64| 76.67| 74.95| 74.85| 45.43| 10   |
>
>
> [R1] Chiang, Ping-yeh, et al. "Loss landscapes are all you need: Neural network generalization can be explained without the implicit bias of gradient descent." ICLR'23.
>
> [R2] Liu, Sijia, et al. "Zeroth-order stochastic variance reduction for nonconvex optimization." Advances in Neural Information Processing Systems 31 (2018).

---

> > ### Comment · Reviewer_X34v · 2023-11-19
> > **Thank you for the clarifications**
> >
> > Thank you for taking your time to add the ablation study on the smoothing parameter and sparsity level. I believe this has improved the quality of the paper even further. As for the first point, I understand that the direct comparison between ZO and FO serves as an upper bound and also demonstrates the scalability merit, from the perspective of ZO methods. However, if you look at it from the perspective of FO practitioners, it seems a little unfair that you only apply sparsity on the ZO method, since pruning and optimization are orthogonal approaches. Note that I am quite positive overall about the quality and significance of this paper. I just don't want it to be unfairly criticized by readers who were distracted from the actual contributions of this paper.

---

> > > ### Author Response · Authors · 2023-11-19
> > > **Thank you for the very positive feedback and insightful suggestion!**
> > >
> > > First, we sincerely appreciate your acknowledgment of our efforts in responding to your comments and for recognizing the enhanced quality of our paper following the rebuttal process.
> > >
> > > Second, we are grateful for your understanding of our intention to compare ZO methods with FO methods from the ZO perspective.
> > >
> > > Lastly, your input from the perspective of FO practitioners is quite valuable to us. We now better understand your concerns regarding FO methods, especially given the differences between pruning and optimization. To alleviate this concern, we would like to highlight that, as depicted in Figure 4, we have also included comparisons between ZO methods and sparsity-induced variants of FO methods. In the revision, we are committed to further enhancing our paper's clarity by providing a comprehensive context for the comparisons between ZO and FO methods. Your feedback is greatly appreciated, and we are dedicated to making these improvements.

---

> ### Author Response · Authors · 2023-11-16
> **Response to Reviewer X34v (Part II)**
>
> **Q1: Rationale behind the comparison of CGE vs. RGE for q=d.**
>
> **A3:** Thanks for raising this question.
> The reason for comparing CGE with RGE at $q = d$ is below.
> + As mentioned in **Sec.3**, the variance of RGE reduced and approached that of CGE. Thus, supposing we use a sufficiently large number of random samples, we wonder about the performance of RGE vs. CGE. With this study, it will help us to  determine which ZO gradient estimator can largely preserve the model accuracy.   This is why we only conduct RGE at $q=d$. In response to the reviewer's suggestion, we have incorporated an additional experiment **Table R2** to showcase the accuracy of using RGE at different values of $q$, as opposed to our setting of $q = d$. As we can see, a larger value of $q$ improves the optimization accuracy. **This is not surprising as shown in [R3], it is established that increasing $q$ leads to a reduction in the convergence error on the order of $\sqrt{1/{q}}$.**
> + In addition to optimization accuracy, we also observed that  **RGE underperforms CGE in computational efficiency** (see **Tab. A1**). Hence, we selected CGE as the base ZO gradient estimator in our proposal.
>
> We agree with the reviewer that “how you can increase the accuracy of estimating the gradient when q << d” is a critical question. In fact, the proposed  sparsity-induced CGE is such an alternative solution.
>
> **Table R2:** Ablation study on query number (q) of RGE when training the on Multi-layer CNN model with 6,670 parameters on CIFAR-10, with the same setup as **Fig. 2**.
> | Query Number ($q$) | 6,670 | 4,000 | 2,000 | 1,000 | 500 | 250 | 100 | 50 | 25 | 10 | 5 | 1 |
> |------------------|-------|-------|-------|-------|-----|-----|-----|----|----|----|---|---|
> | Acc (%)          | 65.31 | 55.13 | 55.11 | 53.15 | 52.10| 48.89|46.61|46.58|42.75|38.03|35.14|25.68|
>
>
>
> **Q2: Why not use automatic differentiation for the modules that are differentiable?**
>
> **A4:** According to the chain rule, the gradient for the whole system can be the product of the FO gradient of the differentiable part and the ZO estimated gradient of the non-differentiable part.
> + **Nevertheless, the application of the chain rule in this scenario can potentially exacerbate the variance of ZO gradient estimation, especially when the FO gradient entries exhibit large magnitudes.** Consequently, this can hinder optimization convergence, particularly during the initial stages of training. In our black-box defense application, our proposed method demonstrates improved performance compared to the baseline method, ZO-AE-DS, which  relies on the chain rule for black-box optimization.
> + Moreover, the application of 'simulation-coupled DL for discretized PDE error correction' further complicates this scenario. This approach involves an n-fold composition of non-differentiable and differentiable modules. **The explicit computation of the gradient in this context, using the chain rule, becomes exceedingly complex and computationally intensive.** In this case, our proposal can bypass this challenge.
>
>
> [R3] Liu, Sijia, et al. "Zeroth-order online alternating direction method of multipliers: Convergence analysis and applications." International Conference on Artificial Intelligence and Statistics. PMLR, 2018.

---

### Official Review · Reviewer_xr4D · 2023-11-03

**Soundness:** 3 good
**Presentation:** 3 good
**Contribution:** 3 good
**Rating:** 6
**Confidence:** 3

**Summary:**

This is the comments from the fast reviewer. The paper presents a well-constructed framework introducing a sparsity-induced, BP-free optimization method for ML tasks that lack explicit first-order information, such as  black-box prompt learning and adversarial attack and defense. The method adeptly balances query efficiency and training effectiveness, outperforming existing techniques while also extending the scalability of zeroth-order optimization.

**Strengths:**

[Strengths]

  -[1] The paper provides a comprehensive and well-designed comparison between CGE and RGE, demonstrate CGE becomes increasingly advantageous as model depth increases.

  -[2] They find one valuable property of CGE is the disentanglement of finite differences across coordinates, which suggests that reducing CGE’s query complexity is aligned with pruning the model weights that are being optimized. And the ZO-GraSP method leverages the flow of gradient signals to determine the sparsity prior of model weights at initialization.

**Weaknesses:**

[Weaknesses]

  -[1] While the paper is generally well-explained, certain components of the proposed method could benefit from further elucidation for increased clarity. For instance, formalizing the derivation process for the layer-wise pruning rate, even if based on a pre-existing method, would significantly enhance the method's understanding and reproducibility. This improvement would fortify the paper's pivotal contribution, ensuring its effective delivery to the audience.

  -[2] In the paper, the introduction of improved inductive biases for ZO deep model training is mentioned, and the authors may need to carefully balance to ensure that these biases do not overly constrain the model's generalization performance. This may involve appropriately controlling the strength of inductive biases during model training to strike a balance between performance on specific tasks and generalization. Considering the advantages and disadvantages of inductive biases in the context of specific application scenarios and tasks is crucial.

**Questions:**

refer to the weakness.
In general I would expect the authors to explain the results:

[1] whether the results in Fig. 4 is SOTA or not. Shall we expect ZO is comparable or better than FO in full weight network? (not only sparse network).

[2] How should we expect ZO v.s. FO with sparsified weights/gradients? do the authors think this is SOTA or not.

[3] How's the ZO applied to other CNNs or even other variants of ResNet beyond resent-20, and resnet-18? This shows that the ZO is not speficially tuned for Resnet-18 or resent-20. It can be applied to various structure/dataset.

---

> ### Author Response · Authors · 2023-11-16
> **Response to Reviewer xr4D (Part I)**
>
> **W1: Further clarification on  Layer-wise Pruning Ratios (LPRs).**
>
> **A1:** According to the findings presented in [R1], the effectiveness of the sparse model predominantly hinges on the layer-wise pruning ratios. Bearing this insight in mind, we employed our proposed ZO-GraSP to effectively find the (binary) pruning mask and then extracted the LPRs $p^{(l)} \%$ from this mask for every layer $l$. Thanks to LPRs, we can then integrate the pruning information into ZO training using dynamic, sparse gradient updates. The resulting optimization framework can also be regarded as a zeroth-order stochastic block-wise coordinate descent, but the gradient sparsity information is determined by ZO-GraSP and its corresponding LPRs.  We will expand upon and explain this process more thoroughly in the revision.
>
>
> **W2: Weakness in inductive biases.**
>
> **A2:** We concur with the observation that gradient sparsity introduces inductive biases. Indeed, there exists a trade-off between sparsity (efficiency) and performance, which we have rigorously assessed through the lens of convergence error in **Appx. A**. As shown  in **Eq. A1**, as the sparsity ratio $p$ increases, the convergence error may also increase. We acknowledge the importance of effectively managing inductive biases, in line with the Reviewer's valuable suggestion. Encouragingly, we observe that the utilization of stochastic sparsity in gradients yields improved generalization compared to static weight sparsity, as demonstrated in **Fig. 4**. In our future work, we intend to develop a dynamic and automated sparsity scheduler to better adjust the impact of inductive bias during ZO training. This is a very insightful comment. Thanks! We will revise our work for better clarity.
>
>
>
> **Q1 & 2: How should expect ZO v.s. FO? Is it SOTA?**
>
> **A3:** We believe the advanced status of our proposed ZO training pipeline (DeepZero) within the realm of gradient-free training schemes, especially for black-box optimization. **While we do not anticipate ZO outperforming FO  methods, where the latter represents a theoretical upper bound for t.** The rationale behind this is that ZO gradient estimation, even when employing CGE, inherently introduces larger variances when compared to stochastic first-order (FO) gradients; see [R2]. Thus, in **Fig. 4**, under the assumption of the same sparsity configuration, it would be expected that FO optimization outperforms ZO optimization, as evidenced by the performance comparison between DeepZero and FO training with Sparse Gradient. However, when we contrast DeepZero with FO training employing Sparse Weight, a noteworthy observation emerges: our performance is comparable to and could even potentially surpass that of the latter. This is attributed to DeepZero's better sparsity-inducing mechanism during training, which outperforms the approach of training a sparse network with a fixed sparsity pattern.
>
> **In the realm of ZO optimization, we have strong confidence that our method stands as the SOTA**, consistently surpassing even BP-free baselines that are built upon computational graphs (as depicted in **Fig. 1D** and outlined in **Tab. 1**). It is worth noting that we achieve superior generalization performance despite training a neural network of smaller size than these BP-free baselines. This also underscores the significance of the inherent accuracy of the proposed DeepZero optimizer in achieving these results, when compared to other methods.
>
> [R1] Su, Jingtong, et al. ”Sanity-checking pruning methods: Random tickets
> can win the jackpot.” Advances in neural information processing systems 33
> (2020): 20390-20401.
>
> [R2] Liu, Sijia, et al. "Zeroth-order stochastic variance reduction for nonconvex optimization." Advances in Neural Information Processing Systems 31 (2018).

---

> ### Author Response · Authors · 2023-11-16
> **Response to Reviewer xr4D (Part II)**
>
> **Q3: ZO training scalability.**
>
> **A4:** **Scalability-wise, training a model as seemingly modest as ResNet-20 within the ZO optimization framework is a notably non-trivial endeavor.** In fact, to the best of our knowledge, there is no existing work that leverages ZO optimization, which relies on function queries, to train a model like ResNet-20 from scratch while achieving comparable accuracy to our results. As explained in **Sec. 1 and 2**, despite the successes of ZO in tackling machine learning problems, its application has historically been confined to relatively small-scale scenarios. When comparing our approach to the gradient-free training baseline presented in ICLR'23 [R3], **the latter is primarily constrained to small-scale applications, such as models like LeNet under conditions of low-sample and few-shot scenarios.** Based on the above discussion, our work provides an important step in upscaling ZO optimization for DL applications, establishing it as the current SOTA in addressing the challenges posed by black-box neural network training, as outlined in **Sec. 6.2**. In these applications, DeepZero has been conducted on different datasets.
>
> Due to current computational resource limitations, we are constrained from undertaking the training of larger models like ResNet-18. This limitation presents a challenging avenue for future research. We will clarify this limitation in the revision.
>
>
> [R3] Chiang, Ping-yeh, et al. "Loss landscapes are all you need: Neural network generalization can be explained without the implicit bias of gradient descent." ICLR'23.

---

> ### Author Response · Authors · 2023-11-22
> **Reminder on follow-up discussion (1 day left before rebuttal ends)**
>
> Dear Reviewer xr4D,
>
> Thank you very much for the dedicated review of our paper. We are fully aware of the commitment and time your review entails. Your efforts are deeply valued by us.
>
> With only 1 day remaining before the conclusion of the discussion phase, we wish to extend a respectful request for your feedback about our responses. Your insights are of immense importance to us, and we eagerly anticipate your updated evaluation. Should you find our responses informative and useful, we would be grateful for your acknowledgment. Furthermore, if you have any further inquiries or require additional clarifications, please don't hesitate to reach out. We are fully committed to providing additional responses during this crucial discussion phase.
>
> Best regards,
>
> Authors

---

### Meta-Review · Area_Chair_v5H7 · 2023-12-09

**Metareview:**

The research paper introduces a novel framework named DeepZero, which significantly advances the scalability of Zeroth-order (ZO) optimization for training deep neural networks (DNNs). ZO optimization is a technique used when first-order (FO) information, like gradients, is difficult or impossible to obtain. Previously, ZO optimization was limited to small-scale machine learning problems due to challenges in scalability and performance.

DeepZero encompasses three key elements to address these challenges:

1) It demonstrates the superiority of coordinate-wise gradient estimation (CGE) over randomized vector-wise gradient estimation (RGE). CGE is shown to improve both training accuracy and computational efficiency.

2) It proposes a sparsity-induced ZO training protocol, which incorporates model pruning methodology at initialization with finite differences to leverage the sparse nature of deep learning, using CGE.

3) It exploits methods for feature reuse and forward parallelization, enhancing the practicality of ZO training implementations.

Some experiments conducted using DeepZero show that it achieves good accuracy on ResNet-20 trained on CIFAR-10, closely approaching the performance of FO training for the first time. Additionally, DeepZero is applied to certified adversarial defense and deep learning-based partial differential equation error correction, improving performance by 10-20% over the current state-of-the-art methods.

Questions and concerns from reviewers are all carefully addressed by the authors in rebuttal, including a low rating but low confidence and a marginal below borderline but no further response after the rebuttal. The majority of the reviewers think the current manuscript good or above the acceptance borderline after the rebuttal. Therefore the paper could be accepted if the authors made all the promised revision in discussions for the final version.

**Justification For Why Not Higher Score:**

There are some concerns about the claimed novelty and experimental details. One such reviewer keeps marginal acceptance after the rebuttals.

**Justification For Why Not Lower Score:**

Two reviewers think the paper good and two reviewers think the paper above the borderline, after the rebuttal period. One negative reviewer has low confidence and another reviewer of slightly below borderline does not response after the rebuttal.

---

### Decision · Program_Chairs · 2024-01-16

Accept (poster)